# Ran promotes membrane targeting and stabilization of RhoA to orchestrate ovarian cancer cell invasion

Kossay Zaoui[1,2], Zied Boudhraa[1,2,5], Paul Khalifé[1,2,5], Euridice Carmona[1,2], Diane Provencher[1,2,3] & Anne-Marie Mes-Masson[1,2,4]

Ran is a nucleocytoplasmic shuttle protein that is involved in cell cycle regulation, nuclear-cytoplasmic transport, and cell transformation. Ran plays an important role in cancer cell survival and cancer progression. Here, we show that, in addition to the nucleocytoplasmic localization of Ran, this GTPase is specifically associated with the plasma membrane/ruffles of ovarian cancer cells. Ran depletion has a drastic effect on RhoA stability and inhibits RhoA localization to the plasma membrane/ruffles and RhoA activity. We further demonstrate that the DEDDDL domain of Ran is required for the interaction with serine 188 of RhoA, which prevents RhoA degradation by the proteasome pathway. Moreover, the knockdown of Ran leads to a reduction of ovarian cancer cell invasion by impairing RhoA signalling. Our findings provide advanced insights into the mode of action of the Ran-RhoA signalling axis and may represent a potential therapeutic avenue for drug development to prevent ovarian tumour metastasis.

[1] Centre de recherche du Centre hospitalier de l'Université de Montréal (CRCHUM), Montreal, Canada. [2] Institut du cancer de Montréal, Montreal, Canada. [3] Division of Gynecologic Oncology, Université de Montréal, Montreal, Canada. [4] Department of Medicine, Université de Montréal, Montreal, Canada. [5]These authors contributed equally: Zied Boudhraa, Paul Khalifé. Correspondence and requests for materials should be addressed to A.-M.M.-M. (email: anne-marie.mes-masson@umontreal.ca)

Epithelial ovarian cancer (EOC) is the deadliest of all female reproductive system cancers worldwide with 140,000 deaths each year[1–3]. The disease being largely asymptomatic, the vast majority of patients are diagnosed at an advanced stage, which is responsible for a poor prognosis[4]. We have demonstrated that the small GTPAse Ran (Ras-related nuclear protein) is strongly associated with EOC progression, poor overall survival, and a high risk of recurrence[5,6]. Ran is a master regulator of nucleocytoplasmic transport[7,8] and mitotic spindle formation, which are necessary for cell proliferation and cell cycle progression[7,9]. Indeed, we have shown that depletion of Ran prevents EOC cell proliferation in vitro and results in EOC tumor growth arrest in vivo[10].

RhoA is one of the most-studied Rho GTPase, it is activated by guanine-nucleotide exchange factors (GEFs) and is inactivated by guanine-nucleotide dissociation inhibitors (GDIs), which prevent its interaction with the plasma membrane (PM), but not necessarily with downstream targets[11]. In addition, the RhoA protein contains a CAAX motif that influences its targeting to specific plasma membrane (PM) microdomains[12]. However, the CAAX-signaled post-translational modification alone is not sufficient to promote full RhoA membrane association that is required for its proper function[13,14]. RhoA GTPase coordinately regulates multiple aspects of tumor cell invasion[15], and its expression is significantly associated with poor tumor differentiation and advanced stages of ovarian cancer[16].

Here, we investigate the mechanism through which Ran modulates ovarian tumor progression. We find that Ran can localize to the PM where it forms a complex with RhoA GTPase, leading to RhoA stabilization and activation. Our findings describe a signaling pathway involving Ran that regulates EOC invasion through RhoA GTPase activity and may lead to alternative therapeutic strategies for ovarian cancer.

## Results

**Ran stabilizes and co-localizes with RhoA.** Ran, a member of the Ras GTPase family, has been demonstrated to control numerous cellular processes of cancer, including cell proliferation and tumor cell invasion/migration associated with a metastatic phenotype[17–19]. We have previously demonstrated that Ran is overexpressed in invasive high-grade serous EOC cells[6]; however, the role of Ran in EOC cell invasion remains unclear. To address this, we examined the effect of Ran depletion by RNA interference (RNAi) in two aggressive EOC cell lines (TOV-112D and TOV-1946) derived in our laboratory[20,21] (Fig. 1a; Supplementary Fig. 1a). Video microscopy analysis revealed that TOV-112D cells with siRNA-mediated knockdown (KD) of Ran elicited reduced spreading and motility while producing long projections that appeared at the trailing end of cells in comparison with control TOV-112D cells (Supplementary Fig. 1b and Movies 1, 2).

This Ran KD-induced phenotype of elongated cells with pronounced tails is similar to the disrupted RhoA signaling phenotype that has been observed in other systems[22–24]. Therefore, experiments were first carried out to examine RhoA protein levels following Ran KD which demonstrated a drastic decrease in RhoA protein levels (Fig. 1b; Supplementary Fig. 1c). We found a similar effect on RhoA protein levels by targeting the 3′-untranslated region of Ran mRNA or its coding region using either siRNA#2 or siRNA#1, respectively (Fig. 1b; Supplementary Fig. 1c). In contrast, RhoC protein levels were not altered (Fig. 1b; Supplementary Fig. 1c), despite its extensive similarity in protein sequence with RhoA[11]. Importantly, re-expression of a RNAi-resistant Ran wild-type (2xGFP-Ran WT, plasmid containing only the coding sequence) rescued RhoA protein levels in Ran KD cells (Fig. 1a, c; Supplementary Fig. 1a, d) and emphasized the

specificity of this response to Ran. Moreover, Ran KD did not alter mRNA levels of RhoA, Rac1, and Cdc42 (Supplementary Fig. 1e), providing further evidence that the effect of Ran on RhoA protein expression was specific and not due to the inhibition of transcription.

Ubiquitination is reported as a major post-translational modification that regulates RhoA protein stability[25]. To determine whether Ran is implicated in the reduced RhoA levels through the ubiquitin proteasome system, Ran KD cells were treated with the proteasome inhibitor MG-132. We found that MG-132 treatment of Ran KD cells rescued RhoA expression (Fig. 1c; Supplementary Fig. 1d). These results suggest that Ran stabilizes RhoA protein by inhibiting its degradation by the proteasome.

RhoA is localized to the cytosol in mammalian cells and has been reported to translocate to the leading edge of migrating cells and at the membrane ruffles upon activation with for example FBS[26]. However, the increase of RhoA protein levels following MG-132 treatment in Ran KD cells does not reflect the activation status of RhoA. To test this, we performed a GTPase activity assay to determine any change in RhoA activity in response to MG-132 treatment. In the absence of Ran, the decrease in RhoA activity is due to low expression levels of the total RhoA protein in TOV-112D cells under these conditions. However, in Ran KD cells treated with the MG-132, the total level of RhoA protein is similar to control cells, but the RhoA activity is significantly diminished (Fig. 1d). We hypothesized that the reduction of RhoA activity may be due to the absence of RhoA localization to the PM, which is required for RhoA function. To examine the effect of Ran on RhoA cellular localization, Ran KD cells were fractionated to separate PM/lamellipodia and cell body-enriched fractions[13]. Analysis of protein lysates confirmed that RhoA protein levels were decreased in the PM/lamellipodia and cell body-enriched fractions of Ran-depleted cells (Fig. 1e; Supplementary Fig. 1f). Interestingly, RhoA was observed only in cell body-enriched fractions of Ran KD cells treated with MG-132 (Fig. 1f), indicating that RhoA localization to the PM cannot occur in the absence of Ran even after proteasome inhibition. Together, these findings demonstrate that Ran specifically controls RhoA stability and localization to the leading edge of migrating cell. We also demonstrated the co-immunoprecipitation of endogenous Ran and RhoA (Fig. 1h) that can be specifically disrupted by RanBP1 overexpression (Fig. 1h). To test whether the association between Ran and RhoA can occur in non-cancer cells, cell lysates from ARPE-19 cells (a human retinal pigment epithelial cell line) were subjected to co-immunoprecipitation with endogenous proteins Ran and RhoA. Unlike TOV-112D cells, no interaction was detected in ARPE-19 cells (Fig. 1i), suggesting that the association of Ran with RhoA appears specific to ovarian cancer cells.

**Ran promotes RhoA recruitment to the plasma membrane.** In addition to Ran's role in nuclear transport, other examples of Ran's involvement in cytoplasmic signaling pathways have recently included endocytic transport[27,28], the regulation of neuronal outgrowth[28,29]. From budding yeast to mammalian epithelia, Ran is frequently associated with polarized activation of the Rho GTPase Cdc42[30,31]. Moreover, Ran also regulates the Arp2/3 complex[32] and ERM (Ezrin/Radixin/Moesin) activation[33], both of which are signal transducers often linked to RhoA GTPase signaling[34,35]. However, these studies of Ran effector functions were largely limited to their structural effects and their role in cancer cell migration/invasion has yet to be elucidated. Since our findings point to a role for Ran in the recruitment of RhoA to the PM, we further examined the subcellular localization

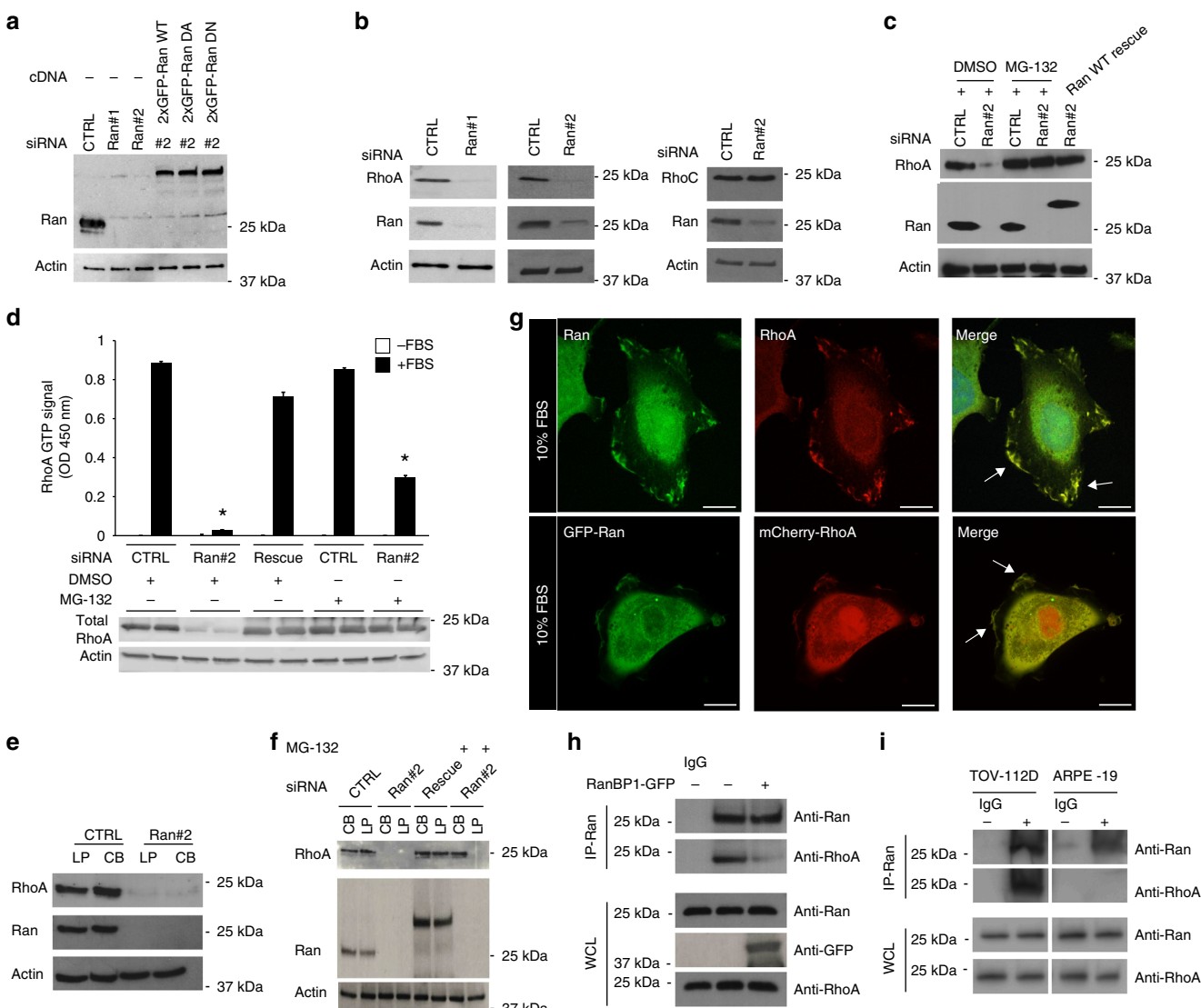

**Fig. 1** Ran GTPase stabilizes and co-localizes with RhoA at the plasma membrane of TOV-112D cells. **a** Western blot of Ran knockdown (KD) with siRNA (CTRL, Ran #1 or 2) and rescue levels with different RNAi-resistant 2xGFP constructs of Ran as wild-type (WT), dominant active (DA), and dominant-negative (DN) in TOV-112D. Actin served as a loading control for all blots. **b** Western blot showing RhoA and RhoC protein expression levels after Ran KD in cells. **c** Western blot showing RhoA protein level after re-expression of 2xGFP-Ran WT (Ran WT rescue) or treatment for 2 h with 20 μM MG-132 in cells transfected with CTRL or Ran #2 siRNA. **d** Active RhoA was examined in cell lysates of control (CTRL), Ran KD or Ran KD with Ran WT rescued. All values are means ± SEM from three independent experiments. *P*-values are based on comparisons with CTRL using the *t* test: *$P < 0.05$ was considered statistically significant. Western blot showing total RhoA. **e**, **f** Cell body (CB) and lamellipodia (LP) of CTRL and Ran KD cells (with or without Ran WT rescued) were fractionated and treated with or without 20 μM MG-132 for 2 h. Equal amounts of proteins were immunoblotted to show RhoA expression in the respective fractions. RhoA was decreased in CB and LP in Ran KD cells, but unchanged in CTRL. RhoA expression is only rescued in CB fractions after treatment with MG-132. **g** Top, TOV-112D cells were fixed, permeabilized, and subjected to immunofluorescence using Ran and RhoA antibodies and DAPI (Merge). Bottom, TOV-112D cells transfected with 2xGFP-Ran and mCherry-RhoA were visualized by spinning disk microscopy. Arrows show Ran and RhoA colocalization at the plasma membrane. **h** TOV-112D cells were transfected with RanBP1-GFP. Protein lysates were subjected to IP with Ran or control IgG antibodies. Proteins were separated by SDS-PAGE and immunoblotted for endogenous RhoA and Ran. **i** Protein lysates from TOV-112D and ARPE-19 cells were subjected to IP with Ran or control IgG antibodies. Proteins were separated by SDS-PAGE and immunoblotted for endogenous RhoA and Ran. Scale bars, 10 μm

and dynamics of Ran in response to serum, which is known to cause Ran activation[36]. We found that Ran is localized predominantly in the nucleus under serum starvation conditions (Supplementary Fig. 1g, h). After 30 min of serum stimulation, Ran was found in the cytoplasm and appeared mainly associated with the nuclear envelope and the PM/ruffles (Supplementary Fig. 1g, h). When cells were stimulated with serum for 1 h, most of Ran re-localizes to the nucleus, but a pool of Ran was still

bound to the PM/ruffles (Supplementary Fig. 1g, h). Microscopy analysis showed that a portion of Ran, both endogenous and exogenous (2xGFP-Ran), colocalized with RhoA to the PM/ruffles (Figs 1g, 2b). Consistent with our previous results in Fig. 1f using MG-132 treatment, RhoA was not able to localize to the PM/ruffles in Ran KD cells (Fig. 2a; Supplementary Fig. 2a, b). However, treatment with MG-132 does not affect the localization of RhoA in control cells (Fig. 2a). Taken together, these

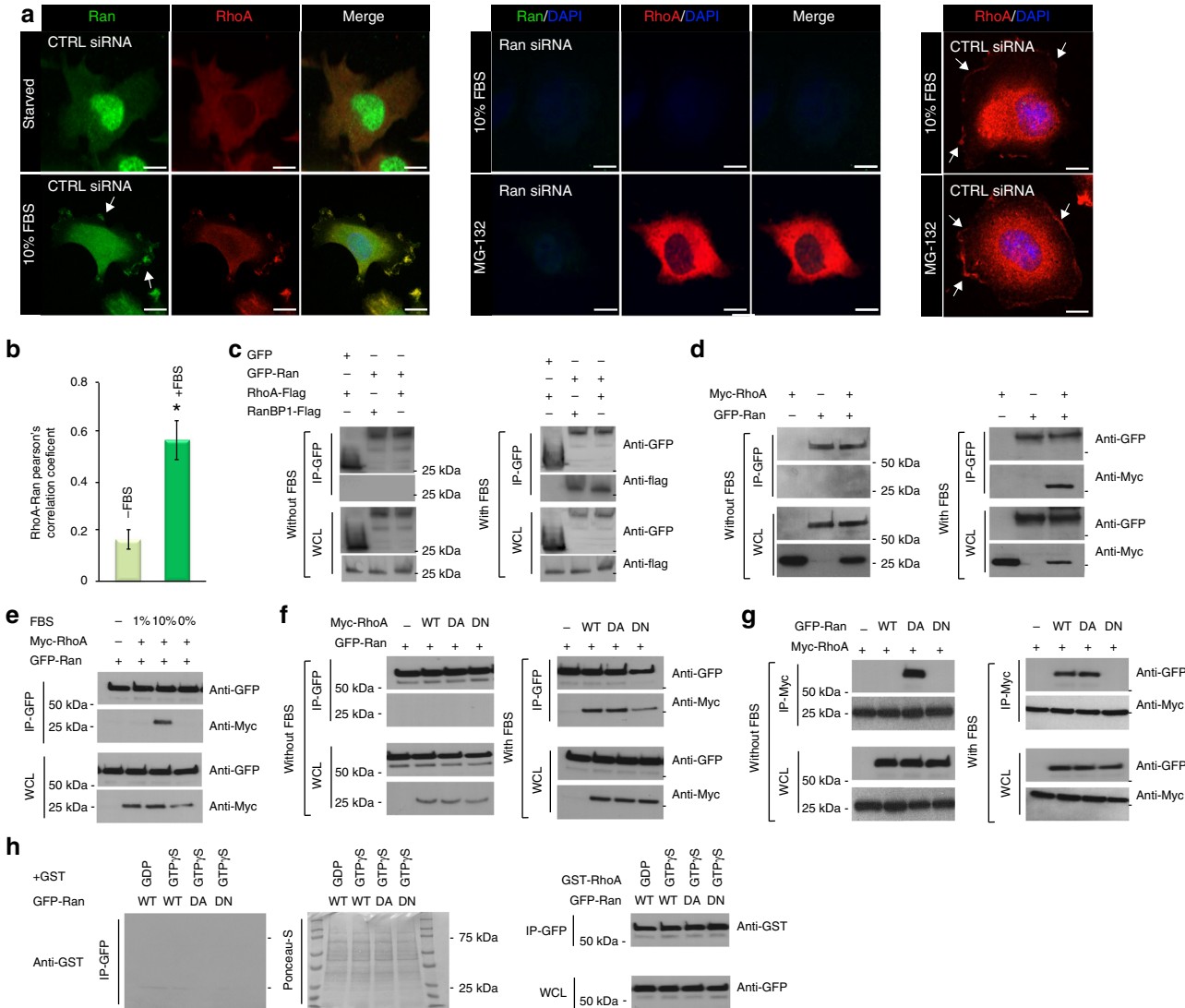

**Fig. 2** Ran GTPase promotes RhoA recruitment to the plasma membrane by direct interaction. **a** TOV-112D cells were either starved or incubated with 10% FBS, treated with or without 20 µM MG-132 for 2 h, and transfected with CTRL or Ran siRNA as indicated. Cells were then fixed, permeabilized, and subjected to immunofluorescence using Ran and RhoA antibodies and DAPI (Merge). Cells were visualized by spinning disk microscopy. Arrows show Ran and RhoA colocalization at the plasma membrane. Scale bars, 10 µm. **b** Colocalization between RhoA (red) and Ran (green) was represented as Pearson's correlation coefficient and measured in individual TOV-112D starved cells or with 10% FBS. All values are means ± SEM from three independent experiments. *P*-values are based on comparisons with CTRL using the *t* test: *$P < 0.05$ was considered statistically significant. **c** TOV-112D cells co-transfected with RanBP1-Flag or RhoA-Flag and GFP alone or 2xGFP-Ran (WT) were starved or incubated with 10% FBS, as indicated. Cell lysates were subjected to immunoprecipitation (IP) with an anti-GFP or an anti-Flag antibody and western blotted as shown. GFP alone was used as a negative control and RanBP1-Flag as a positive control. **d-g** TOV-112D cells co-transfected with Myc-RhoA (WT, DA, or DN) and 2xGFP-Ran (WT, DA, or DN) were starved or incubated with 1% or 10% FBS, as indicated. Cell lysates were subjected to immunoprecipitation (IP) with an anti-GFP or an anti-Myc antibody and western blotted as shown. **h** TOV-112D cells transfected with 2xGFP-Ran (WT, DA or DN), lysed, and subjected to IP with an anti-GFP antibody. Protein complexes were separated by SDS-PAGE and transferred to the nitrocellulose membrane. The membranes were incubated with free GST protein (negative control) or fusion protein GST-RhoA (GDP or GTPγS) and immunoblotted with anti-GST antibody

results confirm that Ran is involved in RhoA localization to the PM/ruffles.

To further determine whether RhoA localization to the PM/ruffles was mediated through Ran, we examined if Ran could associate with RhoA. Ran WT co-immunoprecipitated with RhoA and the positive control RanBP1 only in the presence of 10% serum (Fig. 2c–e), suggesting that this interaction was dependent on both their activation states and their localization. However, no interaction was detected using the GFP empty vector confirming the specificity of Ran interaction with RhoA (Fig. 2c). Consistent

with this interpretation, our results with 10% serum showed RhoA mutants that adopted either a dominant active (DA) or dominant-negative (DN) conformation co-immunoprecipitated with Ran (Fig. 2f). This indicated that Ran transport out of the nucleus was necessary for this interaction, and that Ran re-located to the PM/ruffles as shown in Supplementary Fig. 1g. Similarly, reciprocal co-immunoprecipitations confirmed the ability of the active form (DA) of Ran, which is less concentrated in the nucleus[37], to bind RhoA under serum starvation conditions (Fig. 2g). In the presence of 10% serum, RhoA could efficiently

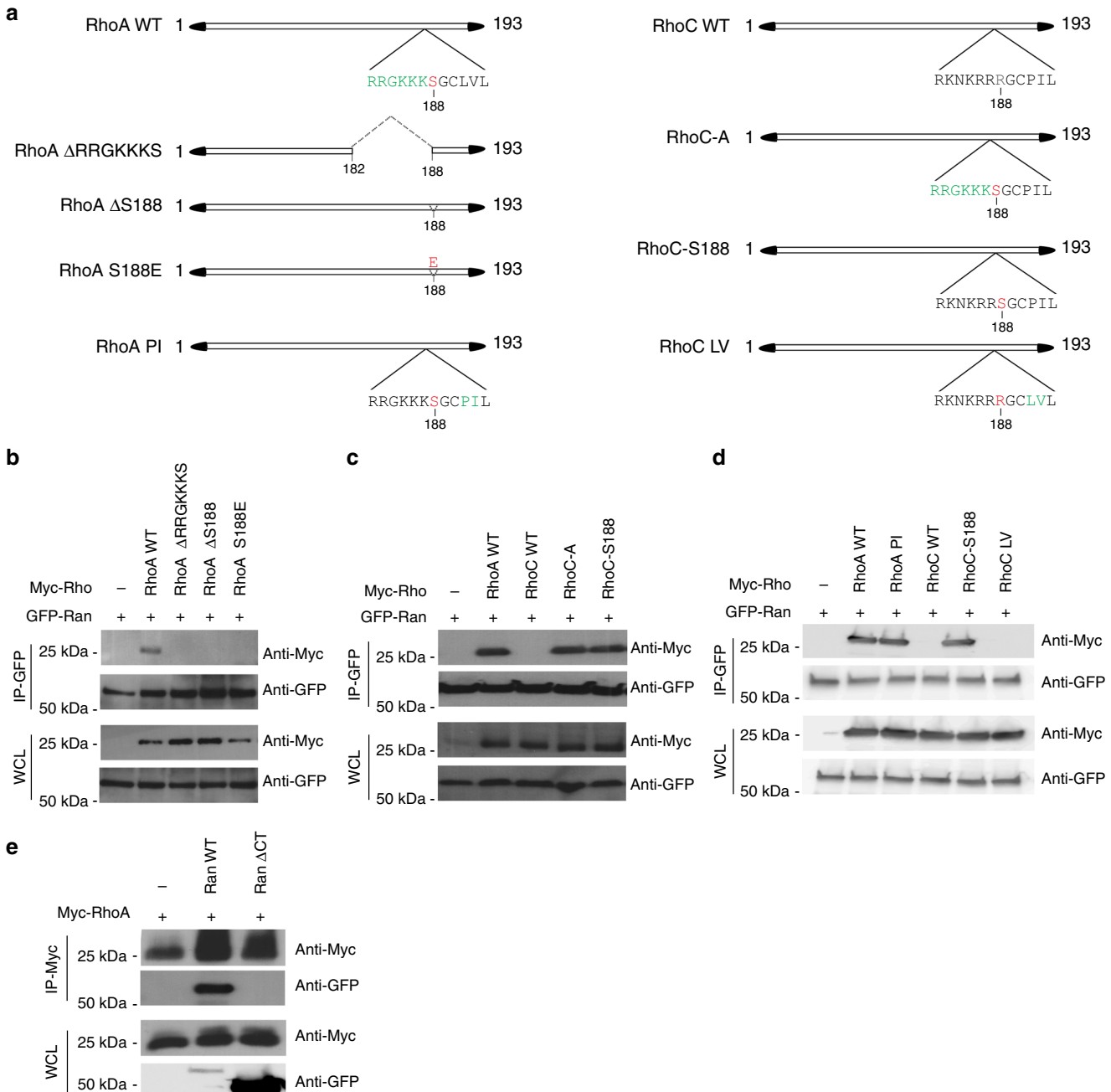

**Fig. 3** The serine 188 of RhoA is required for RhoA interaction with the DEDDDL polyacid domain of Ran. **a** Schematic of RhoA and RhoC mutant constructs. **b** TOV-112D cells were co-transfected with 2xGFP-Ran (WT) and either control, Myc-RhoA (WT), Myc-RhoA (ΔRRGKKKS), Myc-RhoA (ΔS188), or Myc-RhoA (S188E) followed by an immunoprecipitation (IP) using an anti-GFP antibody and western blotted as shown. **c** TOV-112D cells were co-transfected with 2xGFP-Ran (WT) and either control, Myc-RhoA (WT), Myc-RhoC (WT), Myc-RhoC-A, or Myc-RhoC-S188 followed by an IP with an anti-GFP antibody and western blotted as shown. **d** TOV-112D cells were co-transfected with 2xGFP-Ran (WT) and either control, Myc-RhoA (WT), Myc-RhoA (PI), Myc-RhoC (WT), Myc-RhoC-S188, or Myc-RhoC (LV) followed by an IP with an anti-GFP antibody and western blotted as shown. **e** TOV-112D cells were co-transfected with Myc-RhoA (WT) or 2xGFP-Ran (WT) or EGFP-Ran ΔCT (Ran without DEDDDL motif) followed by an IP with an anti-Myc antibody and western blotted as shown

bind to Ran WT and Ran DA but not the dominant-negative form (DN) of Ran, which is localized in the nucleus (Fig. 2g)[37]. To determine whether the interaction between Ran and RhoA was direct, we used far-western blot analysis to examine the ability of Ran WT, Ran DA, and Ran DN to interact with RhoA purified from bacteria as GST-RhoA GDP or GTPγS fusion proteins. Interestingly, RhoA associated with Ran WT, Ran DA, and Ran DN, as detected by anti-GST antibody (Fig. 2h). Taken

together, these data demonstrated an undescribed direct interaction between Ran and RhoA that was dependent on Ran localization but not activity.

**Serine 188 of RhoA is crucial for RhoA and Ran interaction.** Because the C-terminus of RhoA is essential for correct localization of this protein[11], we therefore generated multiple RhoA mutants (Fig. 3a) and performed co-immunoprecipitations to

identify the precise domain motif of RhoA that interacts with Ran. The deletion of the RRGKKKS residues at the C-terminus of RhoA disrupted the interaction with Ran (Fig. 3b). Moreover, the phosphorylation of serine 188 of the RRGKKKS residues protects RhoA from ubiquitin-mediated proteasomal degradation[38], and the removal of serine 188 disrupted the interaction of RhoA with Ran (Fig. 3b). To directly analyse whether the phosphorylation of serine 188 specifically affects RhoA and Ran binding, we performed a co-immunoprecipitation using the serine 188 phosphomimetic RhoA (S188E)[38], which revealed that RhoA S188E failed to co-immunoprecipitate with Ran (Fig. 3b). These results thus provide evidence that the serine 188 of RhoA is crucial for Ran and RhoA interaction.

Although the amino acid sequences of RhoA and RhoC are 88% identical, there exists a major divergence in their C-terminus regions[11]. We demonstrated that RhoC fails to co-immunoprecipitate with Ran (Supplementary Fig. 2c). To confirm the specificity of the serine 188 for the interaction of RhoA with Ran, we generated two mutants of RhoC, where the RRGKKKS amino acids (RhoC-A) or serine 188 alone (RhoC-S188) substituted the amino acids at the corresponding positions to mimic the sequence of RhoA WT (Fig. 3a). We found that both mutants RhoC-A and RhoC-S188 co-immunoprecipitated with Ran (Fig. 3c), confirming that serine 188 is required for RhoA interaction with Ran. To further define the specific role of serine 188, two other mutants where created, RhoA PI and RhoC LV, in which we exchanged between RhoA and RhoC their corresponding amino acids PI and LV in the hypervariable region, downstream S188 (Fig. 3a). RhoA PI displayed a similar interaction with Ran as RhoA WT. However, unlike RhoC-S188, RhoC LV did not bind to Ran (Fig. 3d). Taken together, these results indicate that serine 188 of RhoA is indispensable for the interaction with Ran. According to the role of the carboxyl-terminal DEDDDL domain of Ran in the nucleocytoplasmic transport[39,40], a co-immunoprecipitation using the mutant of Ran without the conserved acidic domain DEDDDL (Ran ΔCT) was performed. We found that, the deletion of DEDDDL motif of Ran perturbs its interaction with RhoA (Fig. 3e), proving that the DEDDDL motif of Ran is required for its interaction with RhoA in a transient/competitive manner.

**Ran recruits RhoA to subcellular structures**. Given that Ran and RhoA colocalized to the PM/ruffles of TOV-112D cells (Fig. 1g) and Ran forms a complex with RhoA (Figs 1h, 2c), we reasoned that Ran could recruit RhoA to the PM/ruffles, allowing spatially restricted activation of RhoA signaling in migrating cancer cells. To explore whether Ran is selectively required for RhoA recruitment to the PM/ruffles, Ran was targeted to a different subcellular membrane, the mitochondria. A fusion chimeric protein was generated (MitoGFP-Ran WT) which colocalized with a mitochondrial probe, MitoTracker® (Supplementary Fig. 2d). Importantly, the mCherry-RhoA WT localized to the mitochondria following co-expression with MitoGFP-Ran WT (Fig. 4a). EGFP-Ran ΔCT (Ran without DEDDDL motif) failed to localize to the PM/ruffles of cells and RhoA remained localized to the cytoplasm (Fig. 4b, c). These results support a role for Ran in recruiting RhoA to the PM/ruffles.

Serine 188 preserves RhoA from proteasome-mediated degradation[38]. To better characterize the Ran/RhoA association, we investigated the subcellular localization of the mutants RhoA ΔRRGKKKS, RhoA S188E and RhoA ΔS188 following their overexpression either alone or with MitoGFP-Ran WT in TOV-112D cells treated with MG-132 to stabilize the expression of these mutants. In contrast to RhoA WT, we found that these mutants do not accumulate to the PM/ruffles in the cells (Fig. 4d, e). Furthermore, in the majority of TOV-112D cells, these mutants do not follow MitoGFP-Ran WT to the mitochondria (Supplementary Fig. 2e, 3a, b). Taken together, these results show that the serine 188 of RhoA and the C-terminus domain of Ran are necessary for their interaction and consequent association with the PM/ruffles.

**Ran-RhoA pathway regulates cell proliferation and invasion**. Despite frequent reports of Ran involvement in invasion and metastasis of tumor cells, little is known about the corresponding molecular mechanism[17,18,41]. Therefore, we explored the effect of Ran-RhoA signaling on the migratory and invasive abilities of EOC cells at the third day post transfection in order to avoid a cell migration/invasion result that is biased by the cell death seen at later time points (see the Methods section). We found that Ran-depleted cells showed decreased migration, where the net velocity of living cells was significantly reduced from 0.23 µm/min to 0.1 µm/min (Fig. 5a). The re-introduction of Ran WT to Ran KD cells rescued cell velocity (Fig. 5a). Alternatively, the depletion of Ran or RhoA significantly reduced cell invasion, and the re-introduction of Ran WT to the corresponding Ran KD cells rescued this altered cell invasion (Fig. 5b). However, the expression of Ran ΔCT in Ran KD cells did not restore cell invasion. Contrary to RhoA KD cells that expressed constructs of RhoA WT, RhoC-A, and RhoC-S188, the RhoA KD cells that expressed RhoC WT and RhoA mutants were not rescued and remained attenuated in EOC cell invasion (Fig. 5b). To our knowledge, this is the first report demonstrating the relationship between Ran and RhoA signaling to control EOC cell invasion. Next, we used an encoded red fluorescent protein called KillerRed-membrane fusion that is activated under appropriate light excitation to efficiently kill cells and selectively disrupt protein–protein interactions at the PM[42]. As a complementary approach, GFP-RhoA WT and a generated chimera of Ran fused with KillerRed-membrane (Ran-KillerRed) were transiently expressed in Ran KD TOV-112D cells to exclusively target Ran to the PM, confirming the role of Ran in the recruitment of RhoA to the PM/ruffle and the effect on cancer cell proliferation and invasion.

Intracellular localization of the GFP-RhoA signal was monitored before and after light inactivation of Ran-KillerRed using spinning disk microscopy. As expected, GFP-RhoA accumulated constitutively with Ran-KillerRed to the PM/ruffles (Fig. 5c). Ran-KillerRed inactivation drastically affected RhoA association with the PM/ruffles (Fig. 5c), and consequently, disrupted Ran binding with RhoA (Supplementary Fig. 3c). However, no change in GFP signal distribution from the plasma membrane was observed in cells expressing RhoA-CCKVL, which is a fusion protein containing wild-type RhoA and the palmitoylation motif of RhoB that promotes RhoA constitutive membrane localization (Fig. 5c)[13].

In order to highlight the importance of Ran association to the PM with RhoA signaling on cell proliferation and invasion, we carried out a TOV-112D cell proliferation assay and transwell-invasion assay before and after inactivation of KillerRed alone, Ran KillerRed and RhoA KillerRed (Fig. 5d, e). The cell invasion was tested independently of any effect on cell proliferation as described in the Methods section, and we found that Ran KillerRed or RhoA KillerRed inactivation resulted in a more pronounced inhibition of TOV-112D cell proliferation and invasion compared with KillerRed vector alone (Fig. 5d, e), although a more marked effect was observed on cell invasion than proliferation. These results underline that the role of Ran on

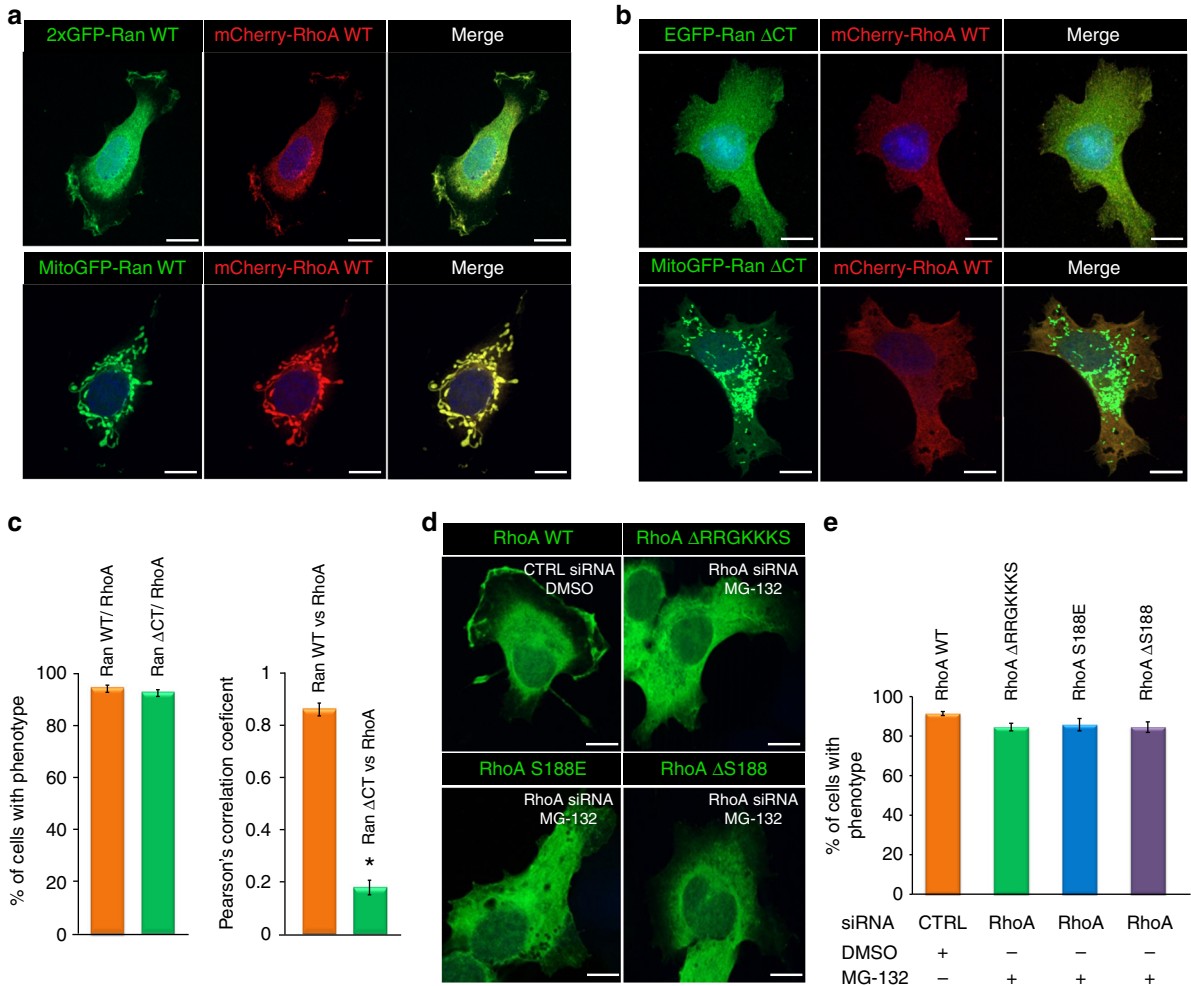

**Fig. 4** Ran GTPase recruits RhoA to subcellular structures. TOV-112D cells expressing mCherry-RhoA WT were co-transfected with 2xGFP-Ran or MitoGFP-Ran WT (**a**), EGFP-Ran ΔCT (Ran without DEDDDL motif), or MitoGFP-Ran ΔCT (Ran without DEDDDL motif) (**b**). Cells were visualized by spinning disk microscopy to establish the localization of RhoA with respect to MitoGFP-Ran or MitoGFP-Ran ΔCT (Ran without DEDDDL motif). Scale bars, 10 μm. **c** Left, percentage of TOV-112D cells with the corresponding phenotype as in (**a**, **b**) for RanWT/RhoA or Ran ΔCT/RhoA colocalization or not to the mitochondria was scored. Right, colocalization between RhoA (red) and Ran (green) represented as Pearson's correlation coefficient and measured for individual TOV-112D cells. All values are means ± SEM from three independent experiments. *P*-values are based on comparisons with CTRL (Ran WT vs RhoA): using the *t* test: **P* < 0.05 was considered statistically significant. **d** TOV-112D cells co-transfected with CTRL or RhoA siRNA and EGFP constructs of either RhoA (WT), RhoA ΔRRGKKS, RhoA S188E or RhoA ΔS188 treated for 2 h with 20 μM MG-132 as indicated. Cells were visualized by spinning disk microscopy. Scale bars, 10 μm. **e** Percentage of TOV-112D cells with corresponding phenotype as in (**d**) for RhoA localization at the PM or not, was scored

ovarian cancer cell invasion, and to a lesser extent cell proliferation, is dependent on RhoA localization/signaling to the PM.

## Discussion

Ran, a member of the Ras GTPase family, has been shown to activate several cancer signaling pathways[10,41]. In this study, we have identified an original role of Ran in the vicinity of the PM to control tumor cell invasion by functionally and specifically linking it to RhoA signaling. This discovery sheds different light on the role of Ran in the fidelity of cell growing and metastasis formation in ovarian cancer. We demonstrate here that down-regulation of Ran affects ovarian cancer cell proliferation and invasion through a proteasome-mediated degradation of RhoA which leads to PM restricted RhoA activity. Ran is a plurifunctional protein and here we show for the first time its PM/ruffles localization. As shown in our model (Supplementary Fig. 3d), we have identified an original role of Ran in association with the PM

to control tumor cell invasion by functionally and specifically linking it to RhoA signaling. Interestingly, it has been reported that Ran is distant from neuronal nuclei and is found in association with the microtubule motor dynein[43]. These findings suggest a mechanism where Ran protein could play a role in microtubule-dependent cellular functions, such as membrane vesicle transport between the intracellular compartments, including the plasma membrane and the nucleus. Moreover, it has been shown that Ran can be secreted and distributed between cells thereby contributing to a localization of Ran to the plasma membrane[44]. Given the ability of Ran to move from cell to cell and its association with microtubules cytoskeleton elements, it is tempting to speculate that an intracellular transport of cargoes loaded with Ran destined for secretion potentially occurs through the export complex. One exciting possibility, although speculative, is that this long-range trafficking of Ran could be a mechanism to explain why a fraction of Ran localizes to the plasma membrane. However, this hypothesis requires further study.

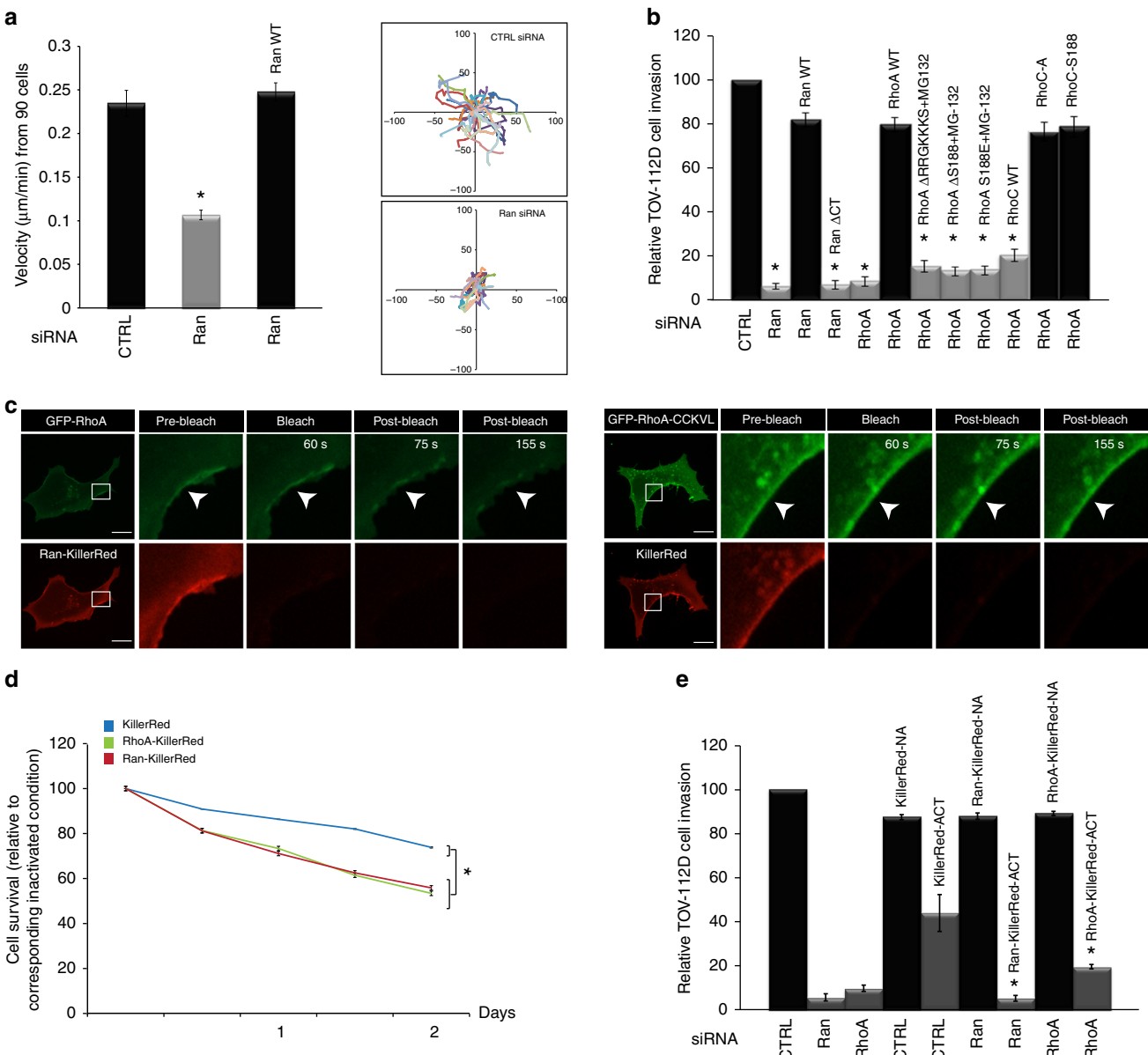

**Fig. 5** Ran regulates cell proliferation and migration/invasion through RhoA recruitment. **a** TOV-112D cells were transfected with siRNAs, and Ran WT as indicated for cell migration assays. Left, cell velocity was determined by tracking living cells. Right, analysis of cell migration paths in CTRL and Ran KD cells. The data represent the trajectories of 30 cells. All values are means ± SEM from three independent experiments. *P*-values are based on comparisons with CTRL using the *t* test: *$P < 0.05$ was considered statistically significant. **b** Effect of Ran-RhoA signaling with or without MG-132 treatment on transwell cell invasion. The invading TOV-112D cells passed through the membrane and were fixed, stained, quantified as described in the Methods section. All values are means ± SEM from three independent experiments. *P*-values are based on comparisons with CTRL using the *t* test: *$P < 0.05$ was considered statistically significant. **c** Left, TOV-112D cells co-expressing Ran-KillerRed and GFP-RhoA were irradiated with green light for 60 s. The illumination resulted in considerable decrease in GFP membrane signal (arrowheads) confirming RhoA detachment from the plasma membrane after light-induced damage of Ran. Right, control experiment showing TOV-112D cells co-expressing KillerRed and GFP-RhoA-CCKVL were irradiated with green light for 60 s. No change in GFP signal distribution from the plasma membrane was observed. Scale bars, 10 μm. **d** Graph shows TOV-112D cell proliferation plotted over time (from the third day post transfection) for each condition as indicated and normalized with corresponding inactivated condition. Values (means ± SEM) from three independent experiments are shown as ratio change in cell survival. *P*-values are based on comparisons with CTRL using the *t* test: *$P < 0.05$ was considered statistically significant. **e** Transwell-invasion assay using transwell chamber before and after KillerRed inactivation. NA non-activated, ACT activated. The data from three independent experiments are expressed as percent change (means ± SEM) compared with the controls. *P*-values based in comparison with KillerRed alone activated conditions using the *t* test: *$P < 0.05$ was considered statistically significant

The existence of RhoA in the nucleus has been reported where it is implicated in regulating the transcriptional activities of specific genes and in the DNA damage response[45–47]. Nevertheless, we did not detect endogenous RhoA in the nucleus of TOV-112D or TOV-1946 cells, suggesting that the major signaling responses observed in

our study are mainly due to the RhoA translocation to the PM/ ruffles. However, in transfected cells, we do on occasion see RhoA signal in the nucleus, although this is most probably due to an artifact associated with overexpression, which has been observed in other mCherry constructs[48].

Several studies have shown that RhoA protein ubiquitination is a post-translational modification that regulates its stability[25,49–51]. Our result showed that, treatment of TOV-112D and TOV-1946 ovarian cancer cells with proteasome inhibitor MG-132 do not increase RhoA protein level compared with DMSO condition. However, in Ran-depleted cells, MG-132 treatment was able to fully restore RhoA and with similar amount of that of control condition (Fig. 1c; Supplementary Fig. 1d). Based on this observation, Ran protein appears to control RhoA protein stability by shifting the balance to favor RhoA degradation, regardless of ubiquitination machinery modification and proteasome degradation system regulation.

Our results demonstrate that, Ran-RhoA complex formation is mediated by the interaction between DEDDDL domain of Ran and Serine 188 of RhoA to control RhoA recruitment to the plasma membrane/ruffles in migrating cells. The fact that Ran DEDDDL domain is essential for mediating Ran interaction with several proteins including RanBP1 (Ran-binding protein 1)[39,52–54] suggests that Ran and RhoA are in transient/competitive interaction and which can be specifically disrupted by adding an excess of one of the known interactors, similarly to a model where Mog1 competes with RCC1 for Ran binding[55,56].

It has been shown that, phosphorylated RhoA at the Serine 188 deactivates RhoA by increasing its interaction with RhoGDI and translocation of RhoA from the membrane to the cytosol[57]. Our data indicate that, RhoA phosphorylation at Serine 188 is not required for the RhoA interaction with Ran and consequently its localization to the PM. However, it could be envisaged that, upon stimulation, RhoA is released from RhoGDI leading to RhoA interaction with Ran and which would allow its stabilization and promotes its localization to the PM/ruffles.

Regulatory control of RhoA protein stability plays a critical role in RhoA-mediated cellular signaling and biological functions[49,50]. Our results unveil a direct interaction between the Ran C-terminal polyacid region and RhoA C-terminal polybasic region. Moreover, the RhoA serine 188 is required for this association. Our approach of manipulating the subcellular location of Ran has provided strong evidences revealing the spatial requirements of RhoA for its localization, stabilization, and activation. The data presented here are consistent with a model in which the RhoA serine 188 overrides the activation state to control RhoA localization. It has been reported that Memo, an effector of the ErbB2 tyrosine kinase receptor, is necessary for RhoA localization and activation at the PM[13]; comparatively, therefore, we propose that Ran acts as a scaffold to coordinate both spatial and temporal engagement of RhoA with guanine exchange factors (GEFs), required for its GTPase activity.

Following Ran depletion, it is conceivable that the alterations in nuclear–cytoplasmic transport may cause abnormal ovarian cancer cell proliferation and migration/invasion. However, the expression of Ran-KillerRed in Ran-depleted cells appears to exclude this possibility. When endogenous Ran is absent, expression of exogenous Ran-KillerRed directs all of the protein to the plasma membrane, and under these conditions, we note an effect on proliferation and migration and this only in the context where Ran-KillerRed is activated (Fig. 5e).

In summary, this study provides an undescribed link between Ran and RhoA signaling that collectively contributes to enhanced ovarian cancer cell growth and invasiveness. In fact, the Ran association with RhoA prevents its ubiquitin-mediated proteasomal degradation through promoting RhoA localization to the PM and then its activation. The fact that ovarian cancer cell proliferation and invasion can be affected by disrupting the interaction between Ran and RhoA provides a rationale to develop advanced pharmacological compounds to prevent ovarian cancer cell progression. Thus, Ran-RhoA signaling complex may be an effective molecular target for controlling cancer metastasis.

## Methods

**Cell culture, transfection, and plasmid constructs.** ARPE-19 a human retinal pigment epithelial cell line was purchased from ATCC (#CRL2302). The TOV-112D and TOV-1946 ovarian cancer cell lines were, respectively, derived from a high-grade endometrioid tumor and a high-grade serous carcinoma, and were used to downregulate the expression of Ran and RhoA. Both cell lines are known to express high levels of Ran[20,21]. Cells (ARPE-19, TOV-112D, and TOV-1946) were grown in the OSE complete medium (Wisent®) containing 10% fetal bovine serum (FBS; Wisent®), 250 µg/mL amphotericin B and 50 µg/mL gentamicin (Wisent®) at 37 °C and 5% $CO_2$[20,21]. Cells were transfected by nucleofection (Amaxa-Lonza®) with 2 µg of siRNA of either CTRL (D-001810-02, Dharmacon®), Ran#1 (J-010353-06-0050, Dharmacon®), Ran#2 (CTM-278994, Dharmacon® a custom designed siRNA targeting 3′UTR of Ran, containing siRNA sequence: GGGUGAAGCU-GAAUAAAGUUCUACUUU), or RhoA (A-003860-18-0010, Dharmacon®). Transfections were also carried out using the following plasmids: 2xGFP-Ran WT, 2xGFP-Ran DA, and 2xGFP-Ran DN (gift from J. Joseph, National Center for Cell Science, India); GFP-RhoA WT and GFP-RhoA-CCKVL (gift from M. Philips, New York University School of Medicine, USA); GFP vector, RhoA-Flag (gift from M. Park, McGill University, Canada); EGFP-RhoC WT and mCherry-RhoA WT (gift from A. Badache, Aix-Marseille Université, Marseille, France); RanBP1-GFP and RanBP1-Flag (gift from P. Lavia, Istituto di biologia e patologia molecolari, Italy); KillerRed-mem (FP966, Evrogen®); pLYS1-FLAG-MitoGFP-HA (Addgene plasmid 50057); Myc-RhoA (WT, DA, DN), Myc-RhoC WT, Myc-RhoA ΔRRGKKKS, EGFP-RhoA ΔRRGKKKS, Myc-RhoA ΔS188, EGFP-RhoA ΔS188, Myc-RhoA S188E, EGFP-RhoA S188E, Myc-RhoA PI, Myc-RhoC-A, Myc-RhoC-S188, Myc-RhoC LV, EGFP-Ran ΔCT (Ran without DEDDDL motif), MitoGFP-Ran WT, MitoGFP-Ran ΔCT (Ran without DEDDDL motif), Ran WT KillerRed, and RhoA WT KillerRed were created by Bio Basic Canada, Inc.

**Random migration assays.** For cell migration, cells were grown on collagen-coated six-well plates (Costar®) for 48 h and were maintained within a chamber (Climabox, Carl Zeiss, Inc) with 5% (v/v) $CO_2$ at 37 °C. The microscope was driven by the AxioVision LE software (Carl Zeiss, Inc) set at ×20 plan Apo 0.8 NA objective and AxioCam MRm (Carl Zeiss, Inc). The motorized stage advanced to pre-programmed locations and photographs were collected for 24 h at 5 min intervals for time-lapse imaging. Motility parameters of living cells including rates of migration and migration paths were obtained from time-lapse imaging. Means of velocity were calculated using MetaMorph® and Microsoft Excel® software[13]. The movies represent the behavior of cells during a 24 h period starting at 48 h post transfection.

**Transwell-invasion assays.** The cell-invasion experiments were based on the results from our random migration assays (Fig. 5a). Since results showed a substantial decrease in the cell displacement/speed of TOV-112D cells 72 h post transfection with Ran siRNA and showed no significant defects on cell proliferation between control and Ran-depleted cells (Supplementary Fig. 1b and Movies 1, 2), we maintained the same conditions as the migration assays to measure cell invasion in order to avoid any bias associated with cell death. Cells were plated on 8.0 -µm porous polycarbonate Transwell membrane inserts (Costar®) that were coated on the bottom with 25 µg/mL rat tail collagen (Sigma®). The lower chamber contained medium with 10% FBS, while the upper chamber was serum free. Cells were plated 48 h after transfection and allowed to migrate through the pores for 24 h. After 1 day, cotton swabs were used to remove non-invading cells from the upper chamber. Migrating/invading cells were fixed with 100% methanol at room temperature, washed with phosphate buffered saline (PBS) and stained with a solution containing 0.5% methylene blue and 50% methanol. Cells were counted with the Count tool of Adobe Photoshop CC® by photographing the membrane inserts using EVOS FLc Cell Imaging System from Invitrogen® (Thermo Fisher Scientific®) and an objective Plan Apo 1.25 × /0.04.

**Immunoprecipitation and western blot analysis.** Cells were harvested in 1% Triton lysis buffer (150 mM NaCl, 20 mM Tris HCl, 1 mM EDTA, 1 mM EGTA, 1% Triton X-100, 1% deoxycholate, at pH 7.4). All lysis buffers were supplemented with 1 mM phenylmethylsulfonyl fluoride (PMSF), 1 mM sodium vanadate, 1 mM sodium fluoride, 10 µg/ml aprotinin, and 10 µg/ml leupeptin. Samples were resolved by SDS-polyacrylamide gel electrophoresis (SDS-PAGE) and transferred to the nitrocellulose. Membranes were blocked with 5% bovine serum albumin (BSA) and probed as described with appropriate antibodies: anti-Ran (sc-271376) diluted 1:1000 in TBS-Tween buffer, anti-RhoA (sc-418) diluted 1:100 in TBS-Tween buffer, anti-RhoC (sc-393090) diluted 1:100 in TBS-Tween buffer, anti-GST (sc-138) diluted 1:100 in TBS-Tween buffer, and anti-Myc (sc-764) diluted 1:100 in TBS-Tween buffer, from Santa Cruz Biotechnology, Inc.; anti-GFP (11814460001) diluted 1:100 in TBS-Tween buffer, from Roche®; anti-Flag (F3165) diluted 1:100 in TBS-Tween buffer, from Sigma®; KillerRed (AB961), diluted 1:1000 in TBS-Tween

buffer from Evrogen® and anti-actin (ab6276), diluted 1:10000 in TBS-Tween buffer from Abcam®. This was followed by incubation with horseradish peroxidase (HRP)-conjugated secondary antibodies: anti-rabbit (sc-2077) diluted 1:10,000 in TBS-Tween buffer or anti-mouse (sc-2061) diluted 1:10,000 in TBS-Tween buffer from Santa Cruz Biotechnology, Inc, or anti-mouse (D3V2A) diluted 1:1000 in TBS-Tween buffer from Cell Signaling Technology®. All immunoblots were visualized by Amersham ECL from GE Healthcare®. For immunoprecipitations, lysates were incubated overnight with antibody at 4 °C with gentle rotation followed by 1 h incubation with protein A- or G-Sepharose beads. Captured proteins were collected by washing three times in lysis buffers, eluted by boiling in SDS sample buffer, and processed as above for western blotting.

**Far-western blotting**. TOV-112D cells were transiently transfected with the indicated constructs, immunoprecipitated with GFP antibody, separated by SDS-PAGE, and transferred to the nitrocellulose membranes. Membranes were incubated with GST-RhoA GDP or GTPγS (Cytoskeleton®) fusion proteins in lysis buffer (20 mM HEPES pH 7.5, 120 mM NaCl, 2 mM EDTA, 10% glycerol, 1 mM PMSF, 10 mg/mL aprotinin, and 10 mg/mL leupeptin), and bound GST-RhoA (GDP or GTPγS) fusion proteins were detected using an anti-GST antibody. For negative control, membranes were incubated with free GST protein[58] (gift from M. Park, McGill University, Canada).

**Subcellular fractionation of the PM/lamellipodia**. For proteins localized in the lamellipodia, cells were plated on 3.0 -μm porous polycarbonate Transwell membrane inserts (Costar®) that were coated on the bottom with 25 μg/mL rat tail collagen (11179179001 from Roche®). The lower chamber contained medium with 10% FBS, while the upper chamber was serum free. Cells were allowed to extend their lamellipodia through the pores. Cell bodies remaining on the upper surface were removed by scraping and the lamellipodia extending to the lower surface were recovered in lysis buffer[13,59].

**Immunofluorescence microscopy and quantification**. Cells grown on collagen-coated coverslips, treated with 20 μM of DMSO (SHBH6857, Sigma®) or 20 μM of MG-132 (C2211, Sigma®) for 2 h, were fixed in 4% paraformaldehyde at room temperature, permeabilized in 0.2% Triton X-100, and blocked with 5% BSA before the addition of primary antibodies. Primary antibodies used for immunofluorescence were against the following: anti-Ran (sc-271376) diluted 1:50 in TBS-Tween buffer, anti-RhoA (sc-179) diluted 1:50 in TBS-Tween buffer and anti-Myc (sc-764) diluted 1:50 in TBS-Tween buffer, from Santa Cruz Biotechnology, Inc. Secondary antibodies Alexa-Fluor 488 or 546 were obtained from Molecular Probes (Thermo Fisher Scientific®) diluted 1:500 in TBS-Tween buffer. The MitoTracker probe (M7514) diluted 1:2000 in OSE complete medium (Wisent®) to label mitochondria is from Invitrogen®. Cells were mounted with the ProLong Diamond Antifade Mountant with DAPI (P36962) from Molecular Probes (Thermo Fisher Scientific®). Images were recorded with a scanning confocal microscope (ZEISS Axio Observer; Carl Zeiss, Inc.) with a ×100 plan Apo 1.4 NA objective and driven by ZEN LE software (Carl Zeiss, Inc.). The degree of colocalization, expressed as the Pearson's correlation coefficient (proportion of all red intensities that have green components among all red intensities), was assessed by the colocalization analysis function of Imaris software (Bitplane®). The results were logged into Microsoft Excel® for analysis. All values are means ± SEM from three independent experiments.

**Rho GTPase activity assay**. The Rho GTPase activation assay was performed using the G-LISA RhoA absorbance-based activation assay (Cytoskeleton®). Briefly, cells were grown on collagen-coated 96-well plates (Costar®), treated for 2 h with 20 μM of DMSO or 20 μM of MG-132 and incubated at 37 °C. At the end of the incubation period, all cells were washed twice with ice-cold PBS and re-suspended in 65 μl of G-LISA lysis buffer. Protein lysates were transferred to ice-cold 1.5-ml centrifuge tubes and clarified by centrifuging at 10,000 rpm for 2 min. Protein concentrations were determined using the Bradford Protein Assay (Bio-Rad®), and 1.0 mg/ml protein was used for the Rho GTPase activation assay as per manufacturer's recommendations. A 1:50 dilution of the primary antibody and 1:250 dilution of the HRP–conjugated secondary antibody were sufficient to produce a RhoA-specific signal. After antibody and HRP reagent incubation, signals were detected on a Versamax microplate reader at 490 nm (Molecular devices®). Data analysis was performed using Microsoft Excel®.

**Live cell imaging**. Cells were grown on collagen-coated coverslips (35 mm, Ibidi GmbH Germany) for 48 h and positioned on a motorized stage equipped with a scanning confocal microscope (ZEISS Axio Observer; Carl Zeiss, Inc.) set at ×100 plan Apo 1.4 NA objective and an Evolve 512 digital camera (Photometrics®) containing a small transparent environmental chamber (Tokai hit®) that was maintained with 5% (v/v) CO₂ in air at 37 °C. The microscope was driven by ZEN LE software (Carl Zeiss, Inc.).

**Light inactivation**. For chromophore-assisted laser or light inactivation (CALI) experiments, TOV-112D cells co-expressing GFP-RhoA and Ran-KillerRed or

GFP-RhoA-CCKVL and KillerRed empty vector were irradiated for 1 min with green light (×100 plan Apo 1.4 NA objective, 515–560 -nm transmitted light at 18 W/cm2) to bleach KillerRed fluorescence. After bleaching, green fluorescence was recorded every second over a period of 5 min with a scanning confocal microscope (ZEISS Axio Observer; Carl Zeiss, Inc.) with a ×100 plan Apo 1.4 NA objective and driven by ZEN LE software (Carl Zeiss, Inc.).

**IncuCyte cell proliferation phase-contrast imaging assay**. For cell proliferation, 20,000 cells/well were seeded for TOV-112D in 24-well plates. Cells were transfected using the following plasmids: KillerRed-mem empty, RhoA WT in KillerRed-mem, and Ran WT in KillerRed-mem incubated for 48 h. Plates were imaged by phase contrast using the IncuCyte™ Live Cell Imaging System (Essen BioScience®). Frames were captured at 2 h-intervals for 7 days from two separate regions/well using a ×10 objective. Proliferation growth curves were constructed using IncuCyte™ Zoom software. Each experiment was performed in triplicate and repeated three times. The data represent TOV-112D cell proliferation from the fifth day post transfection.

**Cell quantification with the corresponding phenotype**. The scanning confocal microscope (ZEISS Axio Observer; Carl Zeiss, Inc) with a ×100 plan Apo 1.4 NA objective and driven by ZEN LE software (Carl Zeiss, Inc) was used to measure the cell number to corresponding phenotypes from three independent experiments ($n = 100$ individual cells). Percentages were calculated using Microsoft Excel®.

**RT-PCR**. The total RNA from TOV-112D cells was isolated using RNeasy Kit (Qiagen®). The total RNA concentration and purity were measured on a Nano-Drop™ spectrophotometer. RNA was reverse-transcribed using QuantiTect Reverse Transcription Kit (Qiagen®) according to the manufacturer's protocol. cDNA amplification was performed with SYBR Green PCR master mix (Applied Biosystems®) using the StepOnePlus Real-Time PCR system (Applied Biosystem®).

Negative controls were included in all experiments, and actin served as the housekeeping gene. Primers were ordered from Integrated DNA Technologies, Inc:
Ran, forward primer: GGTGGTACTGGAAAAACGACC
reverse primer: CCCAAGGTGGCTACATACTTCT
RhoA, forward primer: AGCCTGTGGAAAGACATGCTT
reverse primer: TCAAACACTGTGGGCACATAC
Cdc 42, forward primer: CCATCGGAATATGTACCGACTG
reverse primer: CTCAGCGGTCGTAATCTGTCA
Rac1, forward primer: ATGTCCGTGCAAAGTGGTATC
reverse primer: CTCGGATCGCTTCGTCAAACA

**Statistics**. All statistical analyses were performed using Microsoft Excel®. Graphed data represent the average values ± SEM from at least three independent experiments. Two-tailed, paired Student's $t$ test was used to determine the statistical significance unless otherwise specified.

**Reporting summary**. Further information on research design is available in the Nature Research Reporting Summary linked to this article.

## Data availability
The authors declare that the data supporting the findings of this study are available within the paper and its Supplementary Information files. If needed, additional information is available from the corresponding author upon reasonable request.

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

## Acknowledgements

We thank members of the Mes-Masson laboratory for their helpful comments on the paper. A.-M.M.-M. and D.P. are members of the Centre de recherche du Centre hospitalier de l'Université de Montréal (CRCHUM), which receives support from the Fonds de recherche du Québec - Santé (FRQS). We thank Drs. J. Joseph, Philips, Badache, Park, Lavia, and Mootha for kindly providing expression constructs. We acknowledge Dr. Aurélie Cleret-Buhot from the imaging facility at CRCHUM for technical assistance. This research was supported by the Institut du cancer de Montréal (ICM) and by the Canadian Institutes of Health Research (CIHR) grants (MOP142724 and PJTI48642) to A.-M.M.-M. and D.P. Ovarian tumor banking was supported by the Banque de tissus et de données of the Réseau de recherche sur le cancer of the FRQS affiliated with the Canadian Tumor Repository Network (CTRNet). Z.B. was supported by a MITACS fellowship.

## Author contributions

K.Z. conceived the project, performed the experiments, and analyzed the data with assistance from P.K. Z.B. performed the RT-PCR experiments. E.C., D.P., and A.-M.M.-M. supervised the study and provided guidance. K.Z. wrote the paper with comments from all authors.

## Additional information

**Competing interests:** The authors declare no competing interests.

