## [Transparent Peer Review File · Nature Communications]

Reviewers' comments:

Reviewer #1, Expertise: ovarian cancer, invasion (Remarks to the Author):

In this manuscript Dr Anne-Marie Mes-Masson and colleagues provide some intriguing observation on the interactions between Ran and RhoA. The work is potentially very insightful and provides new directions regarding the regulation of RhoA. There are some elegant cell biology and biochemistry experiments such as those provided in figure 3. It is, however, disappointing that some basic details are left out of the manuscript as detailed below. There is also insufficient depth in confirming the key observations in figure 1. Here are my concerns:

1- I can find this siRNA "(J-010353-06-0050, Dharmacon®)" on the Dharmacon website but could not find this: "CTM-278994, Dharmacon". The authors do not demonstrate the loss of RhoA with the first siRNA which is rather worrying. They need to clarify the identity of the second siRNA and show its sequence. This is essential to allow readers to blast the sequence and ensure that there is no sequence similarity with RhoA. Why did the authors only use this siRNA to demonstrate the effect? Why not use a few from Dharmacon or others.

2- The rescue experiments are somewhat reassuring. However, there is again a worrying lack of information. The authors say "siRNA resistant" construct was used but do not explain how that construct was made resistant. Is the siRNA targeting the 3' UTR while the plasmid only contains the ORF? Or did the authors induce silent mutations in the ORF that would render the construct resistant?

3- The qPCR results showing that RhoA is not reduced, are important. However, there is no mention in the methods where the primers were obtained from. This needs to be clarified and the primer sequences shown. This is critical for the confirmation of the lack of an off-target effect of the Ran siRNA number 2.

4- The movies are interesting. However, there appears to be a lot of cell death following depletion of Ran as shown by the rounding of many cells. The elongation phenotype could be simply related to cell death.

5- Figures 1G and 2A are not convincing. The image for Ran is highly saturated which indicates that the exposure had to be very high for the peripheral localization of Ran to be seen. Compare the saturation for Ran and the reasonable signal in RhoA. At such high exposure, I am surprised that there is not even a shadow for Ran in the Ran siRNA images in figure 2A. This is a critical piece of data that is important for the conclusion of the manuscript. Much more clarity about the methods used and much better images with no saturation are needed.

6- The co-IP experiments in figure 2c lack important negative and positive controls and as such they are not easy to interpret.

7- The experiments in figure 3 are nicely done particularly the modification of RhoC to elicit co-immunoprecipitation. However, co-IP using endogenous levels will be important to demonstrate that the interaction is not an artefact of overexpression.

8- The effects seen in figures 5a and 5b could be explained by the inhibition of proliferation. The authors need to clarify what controls were used to account for this potential source of bias.

Reviewer #2, Expertise: Rho GTPases (Remarks to the Author):

This manuscript makes the interesting and provocative claim that the Ran GTPase localizes to the plasma membrane at protrusions, in an ovarian cancer cell line (TOV-112D) and is the major mechanism for recruitment to protrusions of the RhoA GTPase. They also show that RhoA levels are strongly reduced by silencing of Ran expression. The interaction of Ran with RhoA appears to be direct, and involves the C-terminus of each GTP binding protein: the DEDDDL sequence in Ran and Ser188 in RhoA. Remarkably, expression of a mitochondria-targeted GFP-Ran recruits all detectable Cherry-RhoA. The authors also use Ran-KillerRed to ablate the Ran and show loss of GFP-Ran at the plasma membrane.

While potentially interesting and important, in my opinion the data as presented do not provide sufficient support for the conclusions of the manuscript. The main problems are outlined below. Major issues:

1. No evidence is provided in the initial figures for an interaction of endogenous Ran and RhoA. Immunoprecipitations are all with over-expressed proteins, which could lead to artefactual interactions. The level of over-expression is not measured but must be very high because no detectable protein is present in the whole cell lysate lanes (e.g., Fig 2) even though Ran contributes ~0.4% of the total cell protein.
2. Also in Fig. 2c, d, a control is needed to demonstrate that RhoA is not sticking nonspecifically to GFP.
3. Figure 2a lacks an inset to show colocalization of GFP-Ran with the RhoA – it is not visible in my copy of the manuscript.
4. The authors argue that Ran stabilizes RhoA protein, based on the fact that mRNA levels do not change, but RhoA protein level is increased by treatment with MC312. However, silencing of Ran will reduce mRNA export from the nucleus, and also interfere with ribosome synthesis and assembly, and this will likely result in less RhoA being synthesized. If it is turning over faster than RhoC, then one would expect the differential effect seen in the data.
5. The data showing colocalization of Ran and RhoA at protrusions (in the presence of 10% FBS; Fig 1g) lacks a control to prove that the apparent accumulation is not simply based on increased thickness of the cytoplasm at the protrusion edge. This is a well-known problem, not usually solved by confocal microscopy, since the z-slices can be thicker than the depth of the spread cell on the plate. A control using soluble Cherry, not fused to another protein, is needed as a control here.
6. I do not understand Figure 1d: if RhoA is depleted by silencing Ran expression, then that would account for the reduced RhoA-GTP signal. The data need to be normalized for RhoA expression level.
7. In Fig. 3 the authors argue that S188 is both necessary and sufficient for binding to Ran. However, since the surrounding sequence of RhoC and RhoA are very similar (very basic residues N-terminal to residue 188, and GlyCys C-terminal to residue 188) they have not proved sufficiency (which anyway seems highly unlikely).
8. The authors offer no explanation or even speculation about why, in the presence of 10% FBS, a small amount of Ran localizes to the plasma membrane. It is also unclear why RhoA is NOT at the plasma membrane, because the C-terminal modifications (methylation, prenylation) promote membrane binding. The data in Figure 4d is not novel – it has been known for many years that phosphorylation of S188 disrupts membrane association.
9. In Figure 5, the authors need to prove that the green illumination to activate KillerRed does not bleach the GFP-RhoA.
10. How do the authors know that the reduced proliferation and invasion caused by silencing of Ran is not the result of global inhibition of nuclear transport?
11. A key issue is whether the observed interaction of Ran and RhoA is general to cells or is specific to ovarian cancer cells. The authors really need to test the association of endogenous proteins in non-cancer cells, to determine the generality of the results.

Referee's Comments to the Authors

Comments to the Author

Reviewer #1, Expertise: ovarian cancer, invasion (Remarks to the Author):

In this manuscript Dr Anne-Marie Mes-Masson and colleagues provide some intriguing observation on the interactions between Ran and RhoA. The work is potentially very insightful and provides new directions regarding the regulation of RhoA. There are come elegant cell biology and biochemistry experiments such as those provided in figure 3. It is, however, disappointing that some basic details are left out of the manuscript as detailed below. There is also insufficient depth in confirming the key observations in figure 1. Here are my concerns:

We thank the reviewer for the critical assessment of our work and for suggesting additional experiments to improve our study.

1- I can find this siRNA "(J-010353-06-0050, Dharmacon®)" on the Dharmacon website but could not find this: "CTM-278994, Dharmacon". The authors do not demonstrate the loss of RhoA with the first siRNA which is rather worrying. They need to clarify the entity of the second siRNA and show its sequence. This is essential to allow readers to blast the sequence and ensure that there is no sequence similarity with RhoA. Why did the authors only use this siRNA to demonstrate the effect? Why not use a few from Dharmacon or others.

The siRNA "CTM-278994, Dharmacon" is a custom siRNA (siRNA Ran#2), and we have now provided additional details in the Methods section (see Cell culture, transfection, and plasmid constructs: page 18, line 11). As shown below in Fig. R1, the effect of Ran depletion on RhoA protein level and the cell migration was measured by also using the siRNA#1: J-010353-06-0050 for which the efficiency is shown in Fig. 1A and Fig. S1A in comparison with (siRNA Ran#2). The specificity of the Ran siRNA effect was controlled by rescue experiments as explained in further detail in our response to comment #2 below. The effect of the siRNA#1 on RhoA protein level and cell velocity is shown in the Fig. R1. Western blots are now added in the Fig. 1B and Fig. S1C.

Fig. R1: Western blot showing RhoA protein level in TOV-112D cells (left) and TOV-1946 cells (right) transfected with CTRL or Ran #1 siRNA after 48 hours. TOV-112D cells (left) and TOV-1946 cells (right) were transfected with Ran #1 siRNA as indicated for cell migration assays and described in Methods section. Cell velocity was determined by tracking 90 living cells.

2- The rescue experiments are somewhat reassuring. However, there is again a worrying lack of information. The authors say “siRNA resistant” construct was used but do not explain how was that construct made resistant. Is the siRNA targeting the 3' UTR while the plasmid only contains the ORF? Or did the authors induce silent mutations in the ORF that would render the construct resistant?

The rescue experiments were performed to avoid any off-target effects of Ran siRNA. Our rescue strategy involved the reintroduction of the Ran cDNA (contains only the ORF) since the siRNA targets Ran on the 3' UTR. This detail has been added to the main text to clarify our approach (see Results section: page 4, line 19).

3- The qPCR results showing that RhoA is not reduced, are important. However, there is no mention in the methods where the primers were obtained from. This need to be clarified and the primer sequences shown. This is critical for the confirmation of the lack of an off-target effect of the Ran siRNA number 2.

The sequences of the primers are now added to the Methods section (see RT PCR: page 24, line 2).

4- The movies are interesting. However, there appears to be a lot of cell death following depletion of Ran as shown by the rounding of many cells. The elongation phenotype could be simply related to cell death.

As we have indicated that RhoA inhibition has been previously reported to cause an extreme elongation of the cells (see Results section: page 4, line 13), biological models have shown that RhoA inhibition associated with an elongated phenotype leads to an increase in cancer cell migration and invasion, independent of cell death^{1, 2}. The movies represented only the last 2 days from a total recording of 5 days post-transfection with Ran siRNA. Since Ran protein controls cell proliferation, it is understandable to see some dying cells at this stage. However, as shown in the Ran KD movie, the majority of the elongated cells did not undergo cell death, but their movement was highly compromised. The aim was to illustrate simultaneously these two phenotypes after Ran depletion, and we now provide movies that more clearly demonstrate these results. The new movies represent the behavior of cells during a 24-hour period, starting 48 hours post-transfection, which is described in the Methods section for our cell migration assays (see Random migration assays: page 19, line 3). Although TOV-112D cells became elongated and had reduced velocity after Ran depletion, they continued to proliferate for at least 72 hours post-transfection with Ran siRNA. During this time interval, the proliferation program seems to be independent from the elongated and migration/invasion phenotype.

5- Figures 1G and 2A are not convincing. The image for Ran is highly saturated which indicates that the exposure had to be very high for the peripheral localization of Ran to be seen. Compare the saturation for Ran and the reasonable signal in RhoA. At such high exposure, I am surprised that there is not even a shadow for Ran in the Ran siRNA images in figure 2A. This is a critical piece of data that is important for the conclusion of the manuscript. Much more clarity about the methods used and much better images with no saturation are needed.

We were especially meticulous in capturing these images: low exposure time and Gain were used precisely to avoid saturation. These images were taken from a Z-stack (Z-stack=66 plans, section thickness: 0.82 μm) to clearly show the double localization of Ran at the nucleus and plasma membranes simultaneously. We have changed Fig. 1G and Fig. 2A to show better staining of Ran. In Fig. 2A, Ran siRNA images were taken using the three channels (Green, Red, and Blue) as shown below in Fig. R2 in the boxed regions (last two images on the right), which were used to perform line scan analysis of fluorescence intensity (Green, Red) versus distance. We maintain that the absence of any green staining for Ran reflects the precision and high efficiency of the siRNA as show in Fig. 1A. It is possible that the PDF conversion of images may have reduced the high definition and diminished any background of Ran. However, by zooming in the images from high resolution files, we can distinguish a green background as shown below (first two images on the left).

Fig. R2: TOV-112D cells were incubated with 10% FBS, treated with 20 μM MG-132 for 2 hours, and transfected with Ran siRNA as indicated. Cells were then fixed, permeabilized, and subjected to immunofluorescence using Ran (green) and RhoA (red) antibodies and DAPI. Cells were visualized by spinning disk microscopy. Scale bars, 10 μm .

6- The co-IP experiments in figure 2c lack important negative and positive controls and as such they are not easy to interpret.

New co-IP experiments using GFP empty plasmid (negative control) and RanBP1 (positive control) have been included in Fig. 2C to confirm the specificity of the interaction between Ran and RhoA. We describe these experiments in page 6, line 21.

7- The experiments in figure 3 are nicely done particularly the modification of RhoC to elicit co-immunoprecipitation. However, co-IP using endogenous levels will be important to demonstrate that the interaction is not an artefact of overexpression.

Co-IP experiments using endogenous Ran and RhoA proteins were presented in Fig. 3E of the first submitted version of the manuscript. We have performed new co-IP experiments with endogenous Ran and RhoA in TOV-112D cells and ARPE-19 cells, a human retinal pigment epithelial cell line, to confirm the specific association with ovarian cancer cells as shown in the Fig. 3G. The results are described on page 8, line 18.

8- The effects seen in figures 5a and 5b could be explained by the inhibition of proliferation. The authors need to clarify what controls were used to account for this potential source of bias.

This is an excellent comment and allows us to clarify our cell invasion experiments, which were based on the same conditions as our random migration assays (Fig. 5A); (see Random migration assays: page 19, line 3). A substantial decrease was observed in the cell displacement/speed of TOV-112D cells 72 hours post-transfection with Ran siRNA. However, we also observed that cells continued to proliferate for at least 72 hours post-transfection with Ran siRNA as well, indicating that the proliferation program was independent from the migration/invasion phenotype. More details have been added to the Results section: page 9, line 23 and to the Methods section (see Transwell invasion assays: page 19, line 13). Moreover, we performed an inverted invasion assay as described previously^{3, 4}. Using Calcein-AM (Invitrogen) staining, a cell-permeant dye used to determine cell viability, the bottom planes of CTRL and Ran siRNA cells showed the same cell density (Fig. R3). This confirmed that at 72 hours post-transfection, the inhibition of cell invasion was not related to inhibition of cell proliferation. The tracking of living cells and cell speed measurements for control and Ran-depleted cells at 72 hours post-transfection have been added in Fig. 5A.

Fig. R3: Modified Boyden chambers with 8 μm pore size membranes were coated with Matrigel (BD) at a final concentration of 2.5 mg/ml for 2 h at 37°C. Inserts were then inverted and 10⁶ cells transfected with CTRL or Ran siRNA were seeded directly onto the opposite surface of the filter and allowed to adhere for 4 h at 37°C and 5% CO₂. After 4 h, filters were inverted, and the top compartments of the chambers were filled with OSE medium plus 10% FBS, and the bottom compartments with serum free OSE medium. Cells were allowed to invade through the matrix for 72 h and were stained with Calcein-AM (Invitrogen), a cell-permeant dye used to determine cell viability, according to the manufacturer's instructions; cells that did not enter the matrix were removed using cotton swabs. Invading cells were visualized by confocal microscopy, and serial confocal sections (z-stacks) were acquired at 10 μm intervals. The fluorescence intensity

reflecting the number of invading cells was then measured for each section using Cell Profiler software⁵, and cells invading the matrix at least 30 μm and beyond were quantified as the percentage of total cell population within the matrix.

Reviewer #2, Expertise: Rho GTPases (Remarks to the Author):

This manuscript makes the interesting and provocative claim that the Ran GTPase localizes to the plasma membrane at protrusions, in an ovarian cancer cell line (TOV-112D) and is the major mechanism for recruitment to protrusions of the RhoA GTPase. They also show that RhoA levels are strongly reduced by silencing of Ran expression. The interaction of Ran with RhoA appears to be direct, and involves the C-terminus of each GTP binding protein: the DEDDDL sequence in Ran and Ser188 in RhoA. Remarkably, expression of a mitochondria-targeted GFP-Ran recruits all detectable Cherry-RhoA. The authors also use Ran-KillerRed to ablate the Ran and show loss of GFP-Ran at the plasma membrane.

While potentially interesting and important, in my opinion the data as presented do not provide sufficient support for the conclusions of the manuscript. The main problems are outlined below.

Major issues:

1. No evidence is provided in the initial figures for an interaction of endogenous Ran and RhoA. Immunoprecipitations are all with over-expressed proteins, which could lead to artefactual interactions. The level of over-expression is not measured but must be very high because no detectable protein is present in the whole cell lysate lanes (e.g., Fig 2) even though Ran contributes ~0.4% of the total cell protein.

We would direct the reviewer to our response to Reviewer 1, point 7.

2. Also in Fig. 2c, d, a control is needed to demonstrate that RhoA is not sticking nonspecifically to GFP.

New co-IP experiments using GFP empty plasmid (negative control) and RanBP1 (positive control) have been added to Fig. 2C to confirm the specificity of the interaction between Ran and RhoA. The results are described on page 6, line 21.

3. Figure 2a lacks an inset to show colocalization of GFP-Ran with the RhoA – it is not visible in my copy of the manuscript.

We have changed Fig 2A to improve visualization with better Ran and RhoA staining, where arrows indicate the colocalized Ran portion with RhoA at the plasma membrane. It is possible that the PDF conversion of images may have reduced the high definition and diminished Ran and RhoA signal and we are confident that the improved figure addresses this issue.

4. The authors argue that Ran stabilizes RhoA protein, based on the fact that mRNA levels do not change, but RhoA protein level is increased by treatment with MC312. However, silencing of Ran will reduce mRNA export from the nucleus, and also interfere with ribosome synthesis and assembly, and this will likely result in less RhoA being synthesized. If it is turning over faster than RhoC, then one would expect the differential effect seen in the data.

This is an excellent comment. Although Ran protein has been reported to be involved in ribosome and mRNA export^{6, 7}, the exact mechanism is not yet well understood⁸. In addition, numerous carriers and Ran-independent transport mechanisms play a role in the transport into and out of the nucleus, including the transport of mRNA⁷. We do not know whether Ran controls RhoA mRNA export; however, our results showed that a short-term treatment with MG-132 (2 hours) was enough to rescue RhoA protein levels in the absence of Ran in comparison to control cells. We also performed an experiment to measure RhoA and RhoC protein stability in control cells that were treated with Cycloheximide (CHX) to block new protein synthesis. As shown below (Fig. R1), a similar reduction in the levels of RhoA and RhoC was observed at 2 hours, suggesting that the degradation rate of RhoA and RhoC protein is equal. Therefore, this data eliminates the possibility of any differential effect on the balance between RhoA and RhoC protein synthesis and degradation.

Fig. R1: Stability of endogenous RhoA and RhoC in TOV-112D cells. Cells were lysed at the indicated time after cycloheximide (CHX) treatment. RhoA, RhoC and actin immunoblots of total cell lysates are shown.

5. The data showing colocalization of Ran and RhoA at protrusions (in the presence of 10% FBS; Fig 1g) lacks a control to prove that the apparent accumulation is not simply based on increased thickness of the cytoplasm at the protrusion edge. This is a well-known problem, not usually solved by confocal microscopy, since the z-slices can be thicker than the depth of the spread cell on the plate. A control using soluble Cherry, not fused to another protein, is needed as a control here.

We agree with this comment, and we have performed a control using the mCherry vector alone and GFP-Ran. As expected, mCherry alone did not localize at the plasma membrane, confirming the specific colocalization between Ran and RhoA as shown in the figure below (Fig. R2).

Fig. R2: TOV-112D cells transfected with 2xGFP-Ran and mCherry empty vector or mCherry-RhoA were visualized by spinning disk microscopy. Boxed region was used to perform line scan analysis of fluorescence intensity versus distance. Low red signal at the plasma membrane was detected in mCherry transfected cells. In contrast, a strong red signal at the plasma membrane was detected in mCherry-RhoA transfected cells. Scale bars, 10 μ m.

6. I do not understand Figure 1d: if RhoA is depleted by silencing Ran expression, then that would account for the reduced RhoA-GTP signal. The data need to be normalized for RhoA expression level.

The aim of this experiment was to show the impact of Ran depletion on RhoA activity after stabilizing RhoA protein levels with MG-132 treatment. We found that Ran-depleted cells treated with MG-132 showed an increase of RhoA protein levels comparable to amounts in control cells, but these levels did not reflect the activation status of RhoA. RhoA activity was still reduced, suggesting a defect in RhoA localization to the plasma membrane where optimal RhoA activity is correlated with its association with the plasma membrane⁹. Western blotting of total RhoA has been added to Fig.1D to confirm our model.

7. In Fig. 3 the authors argue that S188 is both necessary and sufficient for binding to Ran. However, since the surrounding sequence of RhoC and RhoA are very similar (very basic residues N-terminal to residue 188, and GlyCys C-terminal to residue 188) they have not proved sufficiency (which anyway seems highly unlikely).

The reviewer raises a good point and we agree that we did not fully determine the sufficiency of S188 for binding to Ran. To confirm our findings, we performed new co-IP experiments using two new mutants, RhoA PI, RhoC LV, in which we exchanged their corresponding amino acids PI and LV downstream from S188 as shown in the Fig. 3A. As expected, we found that RhoA PI displayed similar interaction with Ran as RhoA WT. However, unlike RhoC-S188, RhoC LV did not bind to Ran (Fig. 3D). Taken together, these results indicate that serine 188 of RhoA is indispensable for the interaction with Ran. We added these results: page 8, line 7 and, we have changed the sentence in the text: page 8, line 7 to state the following: “*confirming that serine 188 is required for RhoA interaction with Ran.*” As well the sentence in the text: page 8, line 12 to state the following: “*Taken together, these results indicate that serine 188 of RhoA is indispensable for the interaction with Ran.*”

8. The authors offer no explanation or even speculation about why, in the presence of 10% FBS, a small amount of Ran localizes to the plasma membrane. It is also unclear why RhoA is NOT at the plasma membrane, because the C-terminal modifications (methylation, prenylation) promote membrane binding. The data in Figure 4d is not novel – it has been known for many years that phosphorylation of S188 disrupts membrane association.

A- The authors offer no explanation or even speculation about why, in the presence of 10% FBS, a small amount of Ran localizes to the plasma membrane: In the first submitted version of our manuscript, we provided a possible explanation for Ran targeting the PM/ruffles in the Discussion, which included the following text on Discussion section: page 11, line 19.

B- It is also unclear why RhoA is NOT at the plasma membrane, because the C-terminal modifications (methylation, prenylation) promote membrane binding: In the introduction, we had described that CAAX processing alone is necessary but not sufficient to target CAAX proteins such as RhoA to the plasma membrane. Indeed, in the absence of any stimulation signal (FBS or growth factors), the prenylated RhoA is constitutively sequestered in the cytosol by its interaction with RhoGDI, which prevents RhoA localization to the plasma membrane. Targeting of RhoA to the plasma membrane requires a release from this RhoGDI cytosolic chaperone that is induced by an appropriate stimulation (FBS or growth factors)^{10, 11}. Consistent with previous studies, starved TOV-112D cells showed that endogenous RhoA is mainly cytoplasmic; however, for cells stimulated with 10% FBS, RhoA localizes at the plasma membrane as mentioned in Figure. 2A.

C- The data in Figure 4d is not novel – it has been known for many years that phosphorylation of S188 disrupts membrane association: We agree with the reviewer that the role of Serine 188 of RhoA for its stability and localization has been well documented, as indicated in our text: page 7, line 16 and page 9, line 12. However, we used this data (fig. 4D,E) as supplementary controls to demonstrate by immunofluorescence the disruption of RhoA by Ran after the deletion of serine 188. We rewrote the corresponding paragraph to better explain the aim of this experiment: page 9, line 12.

9. In Figure 5, the authors need to prove that the green illumination to activate KillerRed does not bleach the GFP-RhoA.

This is an excellent comment. To assess the specificity of GFP-RhoA dissociation from the plasma membrane after the bleaching of Ran-KillerRed, we performed an important control experiment. TOV-112D cells were transfected with KillerRed empty vector and GFP-RhoA-CCKVL vector, (fusion protein of wild-type RhoA and the CCKVL motif of RhoB, which promotes the constitutive membrane localization of RhoA). We found that the green light does not affect GFP-RhoA-CCKVL localization at the plasma membrane after the bleaching of KillerRed empty vector with the green illumination. These images have been added to Fig. 5C and a description of these results is now found on page 10, line 22.

10. How do the authors know that the reduced proliferation and invasion caused by silencing of Ran is not the result of global inhibition of nuclear transport?

We thank the reviewer for these pertinent comments. Following Ran depletion, it is conceivable that the alterations in nuclear-cytoplasmic transport may cause abnormal ovarian cancer cell proliferation and migration/invasion. However, the expression of Ran-KillerRed in Ran depleted cells appears to exclude this possibility. When endogenous Ran is absent, expression of exogenous Ran-KillerRed directs all of the protein to the plasma membrane and under these conditions we note an effect on proliferation and migration and this only in the context where Ran-KillerRed is activated (Fig. 5E). This explanation is now added on page 13, line 20.

11. A key issue is whether the observed interaction of Ran and RhoA is general to cells or is specific to ovarian cancer cells. The authors really need to test the association of endogenous proteins in non-cancer cells, to determine the generality of the results.

We have performed new co-IP experiments with endogenous Ran and RhoA in ARPE-19 cells, a human retinal pigment epithelial cell line, in comparison with TOV-112D cells to confirm the specific association with ovarian cancer cells. We found that in ARPE-19, Ran does not interact with RhoA. This result has been added to Fig. 3G and a description of these results is found on page 8, line 19.

References

1. Vega, F.M., Fruhwirth, G., Ng, T. & Ridley, A.J. RhoA and RhoC have distinct roles in migration and invasion by acting through different targets. *J Cell Biol* **193**, 655-665 (2011).
2. Konigs, V. *et al.* Mouse macrophages completely lacking Rho subfamily GTPases (RhoA, RhoB, and RhoC) have severe lamellipodial retraction defects, but robust chemotactic navigation and altered motility. *J Biol Chem* **289**, 30772-30784 (2014).
3. Caswell, P.T. *et al.* Rab25 associates with alpha5beta1 integrin to promote invasive migration in 3D microenvironments. *Dev Cell* **13**, 496-510 (2007).
4. Rajadurai, C.V. *et al.* 5'-Inositol phosphatase SHIP2 recruits Mena to stabilize invadopodia for cancer cell invasion. *J Cell Biol* **214**, 719-734 (2016).
5. Stoter, M. *et al.* CellProfiler and KNIME: open source tools for high content screening. *Methods Mol Biol* **986**, 105-122 (2013).
6. Moy, T.I. & Silver, P.A. Nuclear export of the small ribosomal subunit requires the ran-GTPase cycle and certain nucleoporins. *Genes Dev* **13**, 2118-2133 (1999).
7. Macara, I.G. Transport into and out of the nucleus. *Microbiol Mol Biol Rev* **65**, 570-594, table of contents (2001).
8. Cole, C.N. & Scarcelli, J.J. Transport of messenger RNA from the nucleus to the cytoplasm. *Curr Opin Cell Biol* **18**, 299-306 (2006).
9. Kurokawa, K. & Matsuda, M. Localized RhoA activation as a requirement for the induction of membrane ruffling. *Mol Biol Cell* **16**, 4294-4303 (2005).
10. Michaelson, D. *et al.* Differential localization of Rho GTPases in live cells: regulation by hypervariable regions and RhoGDI binding. *J Cell Biol* **152**, 111-126 (2001).
11. Michaelson, D. *et al.* Postprenylation CAAX processing is required for proper localization of Ras but not Rho GTPases. *Mol Biol Cell* **16**, 1606-1616 (2005).

Reviewers' comments:

Reviewer #1 (Remarks to the Author):

The authors have fully addressed all my concerns. I am happy with the results and conclusions.

Professor Ahmed Ashour Ahmed

Reviewer #2 (Remarks to the Author):

In this revised manuscript the authors have addressed many of the issues that were raised concerning the original version of the study, and the data are now much more convincing. There are a few remaining points that are still confusing or are over-interpreted, but most can probably be addressed by changes in the text.

- 1) I still believe that it would be much better to show an endogenous co-IP in Figure 1, since it is so central to the entire thesis of the manuscript. This was brought up in the first review.
- 2) In Figure 1g much of the mCherry-RhoA is nuclear but in Figure 2a and 4a it is all cytoplasmic (as also shown by Adamson et al, JCB 1992). Is this an artifact of over-expression?
- 3) The authors now provide a blot for Figure 1d, but it is still rather confusing, because the data in the absence of MG-132 are not interpretable (there is almost no RhoA in the cells).
- 4) Also in Figure 1F the authors argue from RhoA localization in MG-132 treated cells silenced for Ran that Ran is therefore essential for RhoA membrane localization. However, MG-132 has many effects on cells, which might impact RhoA localization independently of Ran, so this conclusion is overstated.
- 4) Figure 2 states that Ran GTPase recruits RhoA by direct interaction – but the overlay blot is missing a negative control so there is no way to know that the GST-RhoA binding is specific. Also, adding GST-beads to the blot is not a suitable negative control – obviously beads would not bind stably to proteins on nitrocellulose. It would be important to add free GST as this negative control.
- 5) My point 8 commented that the authors do not offer an explanation about why or how Ran localizes to the plasma membrane. The authors state that targeting “may occur through the trafficking network and could also be supported by the fact that a Ran transfer take place between cells...” but neither of these points makes much sense. What trafficking network do they mean? Ran cycles in and out of the nucleus but is this what is meant by trafficking? What has this to do with localization to ruffles? Intercellular transfer of proteins also seems to have nothing to do with localization to ruffles.

Reviewers' comments:

Reviewer #1 (Remarks to the Author):

The authors have fully addressed all my concerns. I am happy with the results and conclusions.

We thank the reviewer for the encouraging remarks and for accepting the revisions of our manuscript.

Reviewer #2 (Remarks to the Author):

In this revised manuscript the authors have addressed many of the issues that were raised concerning the original version of the study, and the data are now much more convincing.

There are a few remaining points that are still confusing or are over-interpreted, but most can probably be addressed by changes in the text.

We thank the reviewer for the critical assessment of our work that are meant to improve the manuscript.

1) I still believe that it would be much better to show an endogenous co-IP in Figure 1, since it is so central to the entire thesis of the manuscript. This was brought up in the first review.

We agree with the reviewer on this point. We have thus moved the endogenous IP results from Figure 3F-G to Figure 1H-I, as suggested (see Results section: page 5, line 24).

2) In Figure 1g much of the mCherry-RhoA is nuclear but in Figure 2a and 4a it is all cytoplasmic (as also shown by Adamson et al, JCB 1992). Is this an artifact of over- expression?

The accumulation of mCherry in the nucleus has been widely reported¹⁻³. However, since in our cell lines the endogenous RhoA is mainly cytoplasmic (Figure 1G, top panel, and Figure 2A), we agree with the reviewer that the nuclear localization of mCherry-RhoA seen in Figure 1G, bottom panel, is most probably an artifact in this experiment due to the level of over-expression. We have made this point more clearly in the discussion where we emphasize that endogenous RhoA is exclusively in the cytoplasm, but we have added the comment that nuclear localization is observed in some transfected conditions, probably due to the artifacts associated with overexpression (see Discussion section: page 12, line 9).

3) The authors now provide a blot for Figure 1d, but it is still rather confusing, because the data in the absence of MG-132 are not interpretable (there is almost no RhoA in the cells).

We apologize for the confusion; we have shown that in Ran KD cells the decrease in RhoA activity is due to its low expression levels of total RhoA protein. However, in Ran KD cells treated with the MG-132, levels of RhoA protein are similar to control cells but the RhoA activity is significantly diminished. Therefore, we hypothesized that the reduction of RhoA activity may be due to the absence of RhoA localization to the PM, which is required for RhoA function. We also show that the stabilization of RhoA level after MG-132 treatment in Ran KD cells does not allow RhoA localization to the plasma membrane (Fig.2A). Moreover, MG-132 treatment of control TOV-112D cells does not affect RhoA localization to the plasma membrane (see below Fig. R1 and new Fig. 2A). We conclude that Ran is

necessary for RhoA localization to the plasma or at least it is involved in the mechanism controlling its localization. We have made this point more clearly in our new revised manuscript on (page 5, line 12).

4) Also in Figure 1F the authors argue from RhoA localization in MG-132 treated cells silenced for Ran that Ran is therefore essential for RhoA membrane localization. However, MG-132 has many effects on cells, which might impact RhoA localization independently of Ran, so this conclusion is overstated.

It has been reported that ubiquitinated RhoA is degraded through the proteasome and MG-132 treatment blocks RhoA degradation⁴. In TOV-112D, the presence of the proteasome blocker, MG132 does not affect RhoA localization to the plasma membrane as shown (see below Fig. R1 and new Fig. 2A).

Fig. R1: MG-132 does not affect RhoA localization to the plasma membrane. TOV-112D cells were treated with 20 μ M MG-132 for 2 hours then fixed, permeabilized, and subjected to immunofluorescence using RhoA (red) antibody and DAPI. Cells were visualized by spinning disk microscopy. Arrows show RhoA colocalization at the plasma membrane. Scale bars, 10 μ m.

Thus we have modified the manuscript to incorporate both the new data as well as a sentence in (page 6, line 23).

4) Figure 2 states that Ran GTPase recruits RhoA by direct interaction – but the overlay blot is missing a negative control so there is no way to know that the GST-RhoA binding is specific. Also, adding GST-beads to the blot is not a suitable negative control – obviously beads would not bind stably to proteins on nitrocellulose. It would be important to add free GST as this negative control.

We agree with this comment. We have repeated an incubated with GST alone as a negative control. Methods were changed accordingly to indicate the source of the free GST protein (page 22). The result is now added to Figure 2H.

5) My point 8 commented that the authors do not offer an explanation about why or how Ran localizes to the plasma membrane. The authors state that targeting "may occur through the trafficking network and could also be

supported by the fact that a Ran transfer take place between cells...” but neither of these points makes much sense. What trafficking network do they mean? Ran cycles in and out of the nucleus but is this what is meant by trafficking? What has this to do with localization to ruffles? Intercellular transfer of proteins also seems to have nothing to do with localization to ruffles.

We thank the reviewer for these pertinent comments. The following explanation about how Ran localizes to the plasma membrane is now added on the Discussion section: Page 11, Line 10 in order to clarify our hypothesis.

Interestingly, it has been reported that Ran is distant from neuronal nuclei and is found in association with the microtubule motor dynein⁵. These findings suggest a mechanism where Ran protein could play a role in microtubule-dependent cellular functions such as membrane vesicle transport between the intracellular compartments including the plasma membrane and the nucleus. Moreover, it has been shown that Ran can be secreted and distributed between cells thereby contributing to a localization of Ran to the plasma membrane⁶. Given the ability of Ran to move from cell to cell and its association with microtubules cytoskeleton elements, it is tempting to speculate that an intracellular transport of cargoes loaded with Ran destined for secretion potentially occurs through the export complex. One exciting possibility, although speculative, is that this long-range trafficking of Ran could be a mechanism to explain why a fraction of Ran localizes to the plasma membrane. However, this hypothesis requires further study.

REFERENCES

1. Fritz, J.V. *et al.* Direct Vpr-Vpr interaction in cells monitored by two photon fluorescence correlation spectroscopy and fluorescence lifetime imaging. *Retrovirology* **5**, 87 (2008).
2. Ali, R., Ramadurai, S., Barry, F. & Nasheuer, H.P. Optimizing fluorescent protein expression for quantitative fluorescence microscopy and spectroscopy using herpes simplex thymidine kinase promoter sequences. *FEBS Open Bio* **8**, 1043-1060 (2018).
3. Muller, M. & Demeret, C. CCHCR1 interacts specifically with the E2 protein of human papillomavirus type 16 on a surface overlapping BRD4 binding. *PLoS One* **9**, e92581 (2014).
4. Bryan, B. *et al.* Ubiquitination of RhoA by Smurf1 promotes neurite outgrowth. *FEBS Lett* **579**, 1015-1019 (2005).
5. Yudin, D. *et al.* Localized regulation of axonal RanGTPase controls retrograde injury signaling in peripheral nerve. *Neuron* **59**, 241-252 (2008).
6. Khuperkar, D., Helen, M., Magre, I. & Joseph, J. Inter-cellular transport of ran GTPase. *PLoS One* **10**, e0125506 (2015).

REVIEWERS' COMMENTS:

Reviewer #2 (Remarks to the Author):

The authors have adequately addressed the final points raised in the second review with new data or changes to the text, and I feel that the study is now suitable for publication without further modification.